# Photothermally heated colloidal synthesis of nanoparticles driven by silica-encapsulated plasmonic heat sources

Aritra Biswas [1], Nir Lemcoff [1], Ofir Shelonchik[1], Doron Yesodi[1], Elad Yehezkel[1], Ella Yonit Finestone[1], Alexander Upcher [2] & Yossi Weizmann [1,2,3] ✉

Using photons to drive chemical reactions has become an increasingly important field of chemistry. Plasmonic materials can provide a means to introduce the energy necessary for nucleation and growth of nanoparticles by efficiently converting visible and infrared light to heat. Moreover, the formation of crystalline nanoparticles has yet to be included in the extensive list of plasmonic photothermal processes. Herein, we establish a light-assisted colloidal synthesis of iron oxide, silver, and palladium nanoparticles by utilizing silica-encapsulated gold bipyramids as plasmonic heat sources. Our work shows that the silica surface chemistry and localized thermal hotspot generated by the plasmonic nanoparticles play crucial roles in the formation mechanism, enabling nucleation and growth at temperatures considerably lower than conventional heating. Additionally, the photothermal method is extended to anisotropic geometries and can be applied to obtain intricate assemblies inaccessible otherwise. This study enables photothermally heated nanoparticle synthesis in solution through the plasmonic effect and demonstrates the potential of this methodology.

Plasmonic nanomaterials display phenomena such as local field enhancement, hot electron carriers, and photothermal conversion, all stemming from the resonant oscillations of surface electrons[1]. Consequently, much effort has gone into elucidating the mechanisms governing the plasmonic effects and applying them in different fields. Over the past two decades, the interest in plasmon-mediated light-to-heat conversion has been steadily growing due to applications in photothermal catalysis, imaging, and most notably photothermal therapy[2–6]. Among the plethora of thermoplasmonic materials developed over the years, gold nanoparticles (AuNPs), particularly anisotropic nanorods and bipyramids are the most popular photothermal agents due to their excellent conversion efficiency, biocompatibility, chemical stability, and ease of large-scale synthesis[7–11]. Gold bipyramids (AuBPs) stand out as light-to-heat converters over different morphologies, as they exhibit tunable narrow localized surface plasmon resonance (LSPR) bands and can be synthesized with high yields[12–14]. Although there are numerous studies suggesting the temperature near the surface of photothermal nanoparticles can reach well above 200 °C, most current research focuses on applications requiring much lower temperatures[15]. For example, photothermal cancer therapy, amongst the most prominent applications, involves only a slight temperature increase from the biological 37 °C to a maximum of 50 °C[4,15]. In the emerging field of photothermal catalysis, the range of temperatures is not limited by the circumstances as in the previous case; however, most studies involve organic transformations, e.g., polymerization and carbon cross-coupling, where exceeding 100 °C is not favorable[2,16,17]. In our present work, we aimed to generate the temperatures necessary for colloidal synthesis of nanoparticles utilizing the photothermal activation of AuBPs.

[1]Department of Chemistry, Ben-Gurion University of the Negev, Beer-Sheva 84105, Israel. [2]Ilse Katz Institute for Nanotechnology Science, Ben-Gurion University of the Negev, Beer-Sheva 84105, Israel. [3]Goldman Sonnenfeldt School of Sustainability and Climate Change, Ben-Gurion University of the Negev, Beer-Sheva 84105, Israel. ✉e-mail: yweizmann@bgu.ac.il

There are several well-known methods for nanoparticle synthesis, each providing different advantages and drawbacks[18–22]. The colloidal-chemical approach enables the ability to monitor and control the nucleation and growth processes leading to a wide scope of possible products with reproducible results[18–22]. However, obtaining crystalline nanoparticles with well-defined morphologies and narrow size distributions entails carefully selecting suitable reagents/reaction conditions and, when considering metal oxides, non-aqueous solvents with elevated temperatures (300–380 °C) are often needed[19–21]. Thus, we hypothesized that a colloidal synthesis initiated by the elevated temperatures produced at the surface of plasmonic nanoparticles could provide an energy-efficient, light-assisted methodology to afford high-quality nanostructures. So far, only a handful of attempts to advance the synthesis of inorganic materials by using a photothermal approach exist and are mostly concentrated on developing lithography techniques[15,23–25]. Boyd and coworkers used a laser to locally heat glass surfaces embedded with gold nanoparticles for the patterning of PbO, and $TiO_2$ in the gas phase[26]. In a study by Yeo and coworkers, metal surfaces were utilized to grow ZnO nanowires by a photothermal-induced reaction[27]. Finally, a glass substrate deposited with gold nanoparticles was used by Robert et al. as a heat source for synthesizing indium hydroxide microcrystals in water[28]. However, these and other current studies, lack the product quality and scope needed to establish the photothermal approach as a viable synthetic pathway towards nanoparticles.

Herein, we present a light-assisted colloidal synthesis of nanoparticles by utilizing plasmonic photothermal silica-coated AuBPs ($SiO_2$@AuBP) to introduce the heat necessary for the nucleation and growth of nanoparticles in solution. We applied our methodology to synthesize monodisperse iron oxide nanoparticles (IONPs) of different sizes and shapes at bulk temperatures substantially lower than conventional procedures (difference from 30 up to 100 °C) with higher yield. To investigate its scope, we implemented the photothermal synthesis to palladium (PdNPs) and silver (AgNPs) nanoparticles including silver nanowires (AgNWs). Furthermore, we demonstrate that the presence of the heat-emitting AuBPs in the light-assisted synthesis plays a key role in the nucleation mechanism, resulting in the observed reaction efficiency and the generation of interesting assemblies. In the case of IONPs and AgNPs, a homogeneous assembly of spherical NPs covering the silica surface of the $SiO_2$@AuBPs could be consistently isolated. As for the palladium, self-assemblies consisting of individual PdNPs clustered into ≈45–65 nm spheres, were formed when utilizing our approach, while they could not be observed when performing the synthesis with conventional heating using a standard hotplate. These superstructures, obtained exclusively through photothermal synthesis, are essential tools for coupling the plasmonic photothermal feature with other properties of nanostructures, such as the generation of hot electrons, magnetism, and electromagnetic hotspots[29–33]. Multifunctional particles of this nature hold vast applicative potential in diverse disciplines[29].

## Results and discussion
### Light-assisted photothermal synthesis of IONPs
To successfully develop a light-assisted synthesis of monodisperse iron oxide nanoparticles, we envisioned a system where $SiO_2$@AuBP would serve as light-to-heat nanoconverters (Supplementary Fig. 1)[34]. The double growth of the silica shell provides two crucial benefits, first, shielding the gold bipyramidal structure from major degradation, and second, assisting in their dispersibility in different solvents (Supplementary Figs. 1, 2)[35–37]. To test the photothermal methodology we selected the synthesis of IONP through pyrolysis of corresponding fatty acid salts, as it normally requires elevated temperatures above 250 °C[19–21]. Thus, we designed a synthetic procedure as follows: Iron-oleate and $SiO_2$@AuBPs were added to a solution of oleic acid, oleylamine, and octadecene, taking advantage of the ability of oleylamine

to decrease the decomposition temperature of iron precursors (Fig. 1a, and Methods). To our satisfaction, optimization of precursor concentration, ligand ratio (oleylamine-oleic acid), and reaction time (Supplementary Tables 1–5) led to nanoparticle formation at photothermal temperatures as low as 170–180 °C (Fig. 1). Increasing the photothermal temperature to 200 °C resulted in larger iron oxide structures (≈5 nm at 180 °C and ≈10 nm at 200 °C, Fig. 1b–d). The temperature of the system could be controlled by regulating the reaction's exposure to the plasmonic resonance wavelength of AuBPs (≈850 nm, 15–20 OD) (Fig. 1b, and Supplementary Fig. 3)[38]. Characterization by X-ray diffraction (XRD) determined the phase of the observed IONPs to be γ-$Fe_2O_3$ (Fig. 1e)[39]. The composition and absence of any $Fe^{2+}$ related impurities were confirmed by X-ray photoelectron spectroscopy (XPS) (Fig. 1f, g, and Supplementary Fig. 4)[40,41]. Lastly, UV-Vis absorption spectra showed no overlap with the near-infrared (NIR) light irradiated ruling out significant absorption of IONPs (Fig. 1h).

Motivated by our initial results we sought to gain a better understanding of the role the AuBPs took in the formation of IONPs (Fig. 2). Thus, the nucleation and growth were investigated by performing the synthesis at different photothermal temperatures, ranging from 160 to 200 °C and analyzing the TEM images of the formed IONPs and the $SiO_2$@AuBPs used to heat each reaction (Fig. 2a, b). The obtained IONPs exhibited narrow size distributions, and the light-assisted synthesis enabled high reproducibility. Notably, we observed that a photothermal temperature of 160 °C was sufficient to achieve the decomposition of the iron-oleate precursor into nuclei, which normally requires temperatures above ≈280 °C[39–42]. Elevating the reaction temperature resulted in increasing the size of the nanoparticles from ≈1–2 nm at 160 °C to ≈10 nm for the 200 °C reaction (170 and 180 °C reactions yielded ≈2–3 and ≈4–6 nm nanoparticles respectively, Fig. 2a, see also Supplementary Figs. 5–7). Interestingly, at low photothermal temperatures (160 and 170 °C) a handful of $SiO_2$@AuBPs appeared to be covered with nucleated IONPs after 4 h of reaction, with the sample isolated (see Methods) from the reaction at 160 °C having substantially more IONPs on the silica surface (Fig. 2b). This observation led us to think that there may be an attractive interaction between the iron precursor and the silica shell, resulting in nucleation on the surface of the photothermal nanoparticles. In addition, under the synthesis condition, the tips of AuBPs were found to be degraded (Fig. 2b, and Supplementary Figs. 8–10).

Next, we turned to a time-dependent analysis of the 200 °C photothermal reaction, acquiring TEM images at three intervals (45, 120, and 180 min) of the four-hour reaction (Fig. 2c). The resulting IONPs showed a gradual increase in average size from ≈4 nm at 45 min to ≈6 nm at 3 hours (Fig. 2c, and Supplementary Figs. 11–13). Remarkably, the AuBPs' silica surface at all three intervals was homogeneously decorated with IONPs, forming intricate hybrid structures (IONP@AuBP) (Fig. 2d, and Supplementary Figs. 14–16). Moreover, the images showed that the IONPs attached to the AuBP surface were slightly larger (≈2 nm) compared to those unattached (Supplementary Fig. 13). Further characterization of the IONP@AuBP by EDS elemental mapping and TEM tomography confirmed the homogeneous arrangement of IONPs surrounding the $SiO_2$@AuBPs (Supplementary Figs. 17, 18, and Supplementary Movie 1). Approximating a magnet to a solution of isolated IONP@AuBP and examining the UV–vis absorption spectrum validated that the hybrid structures retained both the magnetic and plasmonic attributes of the parent nanoparticles (Fig. 2g, and Supplementary Fig. 19). IONP@AuBPs could also be isolated from the photothermal reaction at 180 °C but were not observed under any conditions when conducting this reaction with conventional heating, i.e., in presence of AuBPs but without any light treatment or through silane chemistry (Supplementary Figs. 20–23). Similar observations were also noticed utilizing silica-coated gold nanorods ($SiO_2$@AuNRs) and $SiO_2$@$AuBP_{660}$ (photothermal activation at 660 nm light, Supplementary Figs. 24–27). These experiments unequivocally establish

the generality of the current approach in terms of different plasmonic nanostructures and wavelengths of light used. Finally, zeta potential measurements of SiO₂@AuBPs mixed with different amounts of iron oleate indicated a slight charge neutralization of the SiO₂@AuBP surface (Fig. 2h). The formation of these hybrid structures throughout photothermal IONP synthesis and the neutralizing effect of the iron oleate, reinforced the hypothesis for an interaction between the iron precursor and the silica surface. However, the question of why the IONPs dissociated from the AuBP surface during the full four-hour reaction, remained open.

Thus, an FTIR analysis of AuBPs before and subsequent to a full reaction was performed, revealing the presence of oleic acid and possibly also oleylamine over the silica surface (Fig. 2i). Additionally, IONP@AuBP samples from the different time intervals were subjected to ICP-OES elemental analysis, showing that the iron to gold ratio decreased as the reaction progressed (Supplementary Table 6). These results provide evidence for a gradual reaction occurring between the ligands (oleic acid and oleylamine) and silica encapsulation, altering the surface chemistry to afford non-polar

SiO₂@AuBPs. We reasoned that this could be the cause for the slow detachment of IONPs from the SiO₂@AuBP surface at higher photothermal temperature/time[43]. To further verify, we carried out the photothermal IONP synthesis utilizing octadecyltrimethoxysilane functionalized[44] SiO₂@AuBPs, mimicking the non-polar surface present at the end of the reaction (see Methods). In this case, no significant nanoparticle attachment on the AuBP surface was observed, and the obtained IONPs lacked a well-defined morphology (Fig. 2e, f, and Supplementary Fig. 28).

Based on the detailed results, we hypothesized a mechanism for the light-assisted nanoparticle formation (Fig. 2j). Initially, the iron precursors are attracted to the polar surface of the SiO₂@AuBPs creating an increased effective molarity within the thermal hotspot generated by the photothermal source, therefore initiating the nucleation and growth of nanoparticles at low bulk temperatures. As the formation of nanoparticles advances the ligands also begin reacting with the silica surface gradually decreasing its polarity, up to a point where the attraction/stabilization is lost and the nanoparticles are detached into the solution.

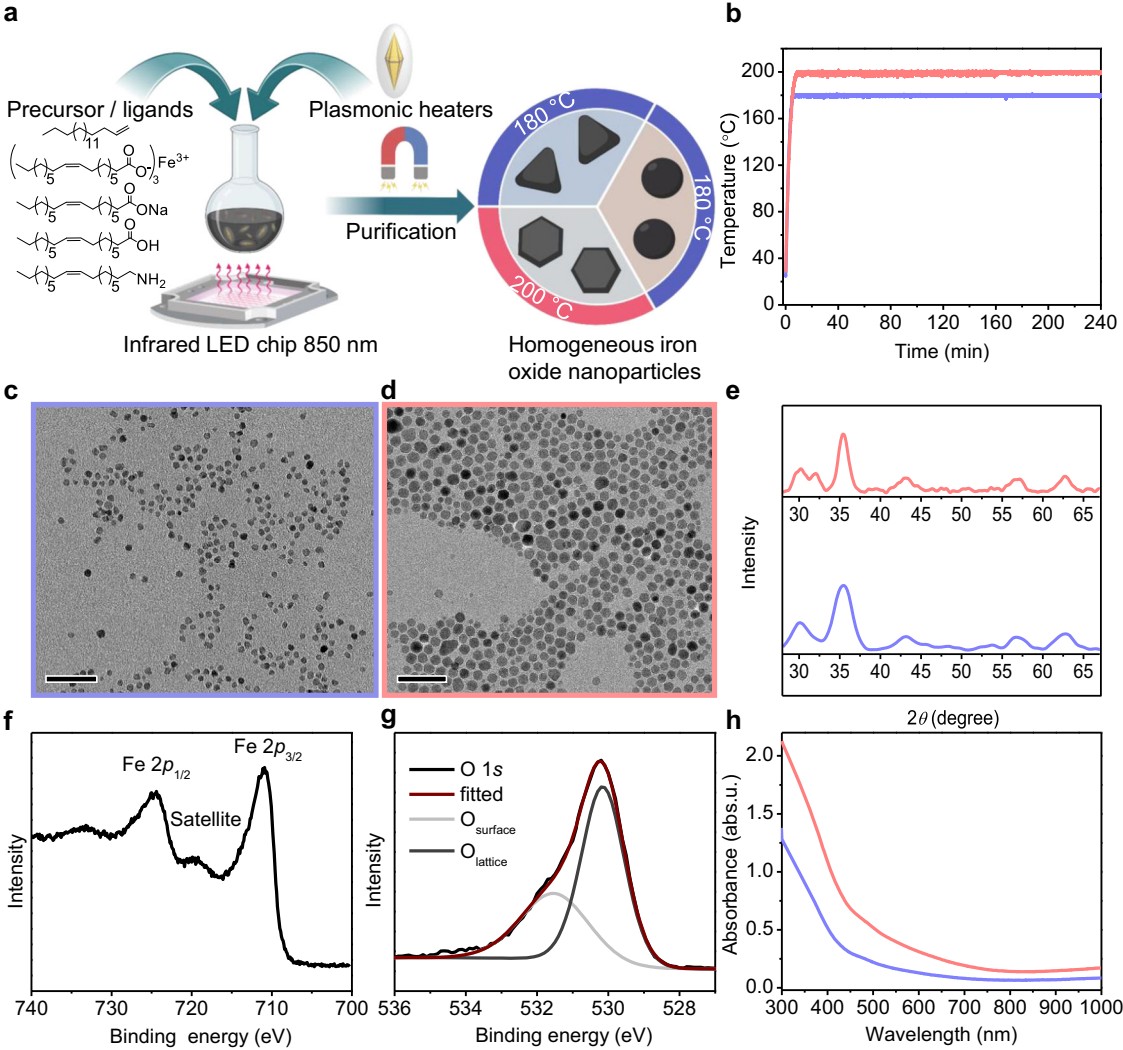

**Fig. 1 | Photothermal synthesis and characterization of iron oxide nanoparticles. a** General Scheme of IONP synthesis. Briefly, iron oleate, oleic acid-oleylamine are mixed with octadecene, and irradiated with 850 nm light in presence of AuBPs (15-20 OD at 850 nm). **b** Temperature profile of two different photothermal reactions (red: 200 °C, blue: 180 °C). **c–d** TEM images of IONPs synthesized under the two photothermal temperatures, the color of the outline corresponds to the temperature profiles in (**b**) (c: ≈4-6 nm at 180 °C, and d: ≈10 nm at 200 °C). Scale bars 50 nm. **e** XRD patterns showing broad features of γ-Fe₂O₃ nanoparticles. Colors again matching to (**b**) and (**c–d**). **f–g** XPS data of the nanoparticles emphasizing the characteristics Fe 2p features of γ-Fe₂O₃ f, and O 1s lattice/surface oxygen associated with nanoparticles and the capping agents g. **h** UV–Visible absorbance spectra of the colloidal iron oxide nanoparticles. The color of the curve matching the sample from (**b–e**). Source data are provided as a Source Data file.

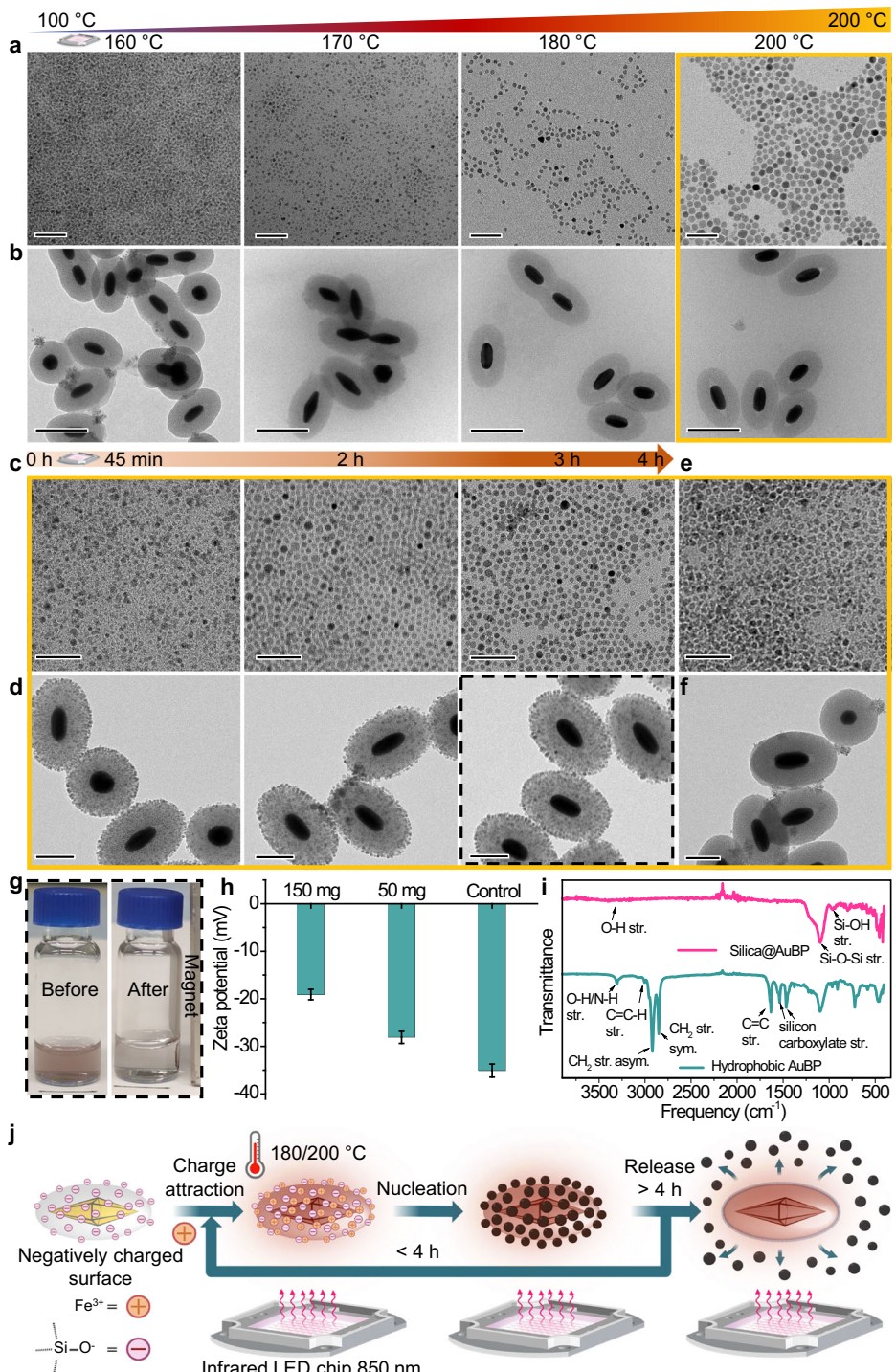

**Fig. 2 | Proposed mechanism of iron oxide nanoparticle formation under photothermal conditions. a–b** TEM images of IONPs a, scale bars 50 nm, and SiO₂@AuBPs b, scale bars 200 nm, isolated post-reaction at different photothermal temperatures (160–200 °C) for 4 h. Nucleation of nanoparticles can be seen at 160 °C, and subsequent growth corresponding to an increase in photothermal temperatures. The yellow color of the outline corresponds to the temperature of 200 °C. **c–d** TEM images of intermediate IONPs c, scale bars 50 nm, and isolated SiO₂@AuBPs d, scale bars 100 nm, at different time intervals (45 min to 3 h) for reactions carried out at a photothermal temperature of 200 °C. A gradual increase in nanoparticle size can be seen as the reaction progressed, as indicated by the brown arrow above. Additionally, IONPs homogenously decorated AuBP encapsulation (IONP@AuBPs). **e–f** TEM images of non-uniform particle formation e, scale

bar 50 nm, in the presence of OTMS-modified SiO₂@AuBPs (at 200 °C, 2 h) and isolated non-polar AuBPs demonstrating poor IONP attachment f, scale bar 100 nm. **g** Digital image of the hybrid IONP@AuBPs (black dotted box) exhibiting a magnetic response when approximating a magnet. Also, the characteristic color of the AuBPs plasmonic interaction can be seen. Scale bar 1 cm. **h** Zeta potential measurements exhibit weak electrostatic interaction between the negatively charged silica surface of AuBP and the iron precursor. Number of measurements ($n = 5$) with standard deviation (SD = 1) as error bar. Control represents absence of iron precursor. **i** FTIR analysis revealing the presence of capping ligands (oleic acid/oleylamine) over the AuBP silica surface after the reaction. **j** Scheme showing the proposed mechanism involved during the photothermal synthesis. Source data are provided as a Source Data file.

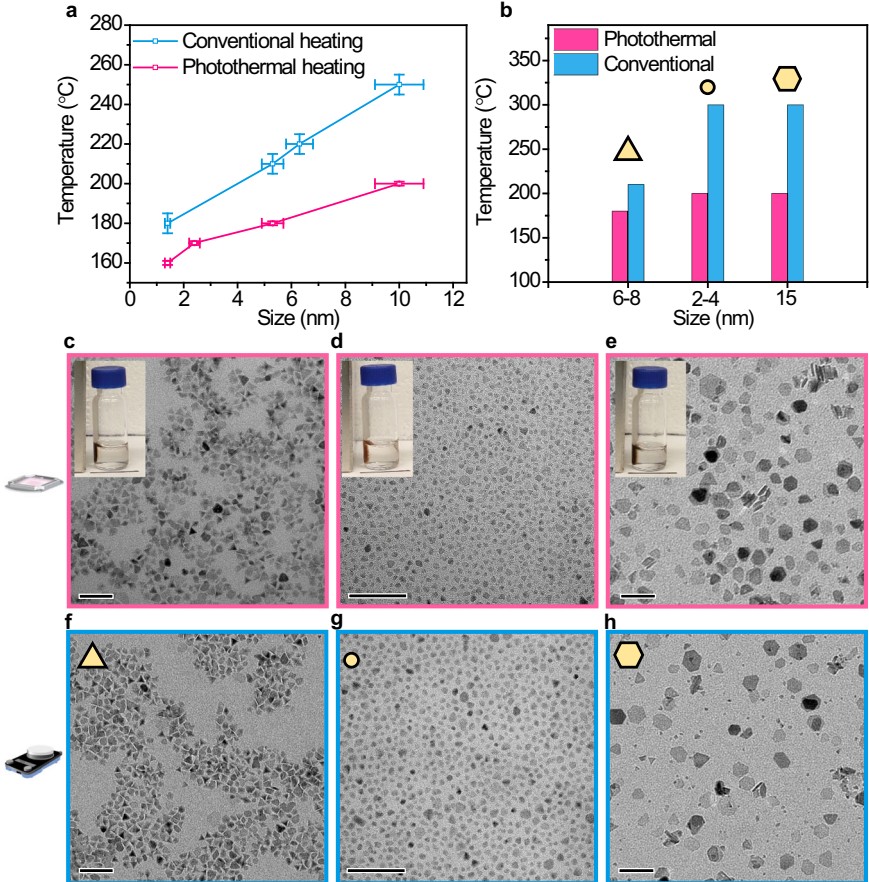

**Fig. 3 | Scope of photothermal approach, and shape modulation.**
**a, b** Comparison of the photothermally synthesized nanoparticle with conventional heating showing a, for a particular size (≈5 nm at 180 °C, $n = 131$; ≈10 nm at 200 °C, $n = 122$ with SD as error bar, as described in Fig. 2a, and Supplementary Fig. 5) or b, shape of nanoparticle (yellow with black outline), photothermal heating is much more efficient compared to conventional heating (≈30–100 °C). **c–e** TEM images of (**c**), triangular iron oxide nanoparticle (≈6–8 nm, synthesized photothermally at 180 °C, as described in Fig. 1a), (**d**), spherical iron oxide nanoparticle (≈2–4 nm; synthesized photothermally at 200 °C, reference [39]), and e, nanoplate of iron oxide (≈15 nm; synthesized photothermally at 200 °C, reference [42]). **c–e** Photothermal heating (red outline), and (**f–h**): conventional heating (blue outline), demonstrating almost intact morphology of the nanoparticles synthesized utilizing photothermal activation at a lower temperature (≈180–200 °C) compared to conventional heating at ≈210–300 °C (**f**, 210 °C, and **g–h**, 300 °C). Scale bars 50 nm. Inset showing the magnetic response of the nanoparticles in the presence of a laboratory-based bar magnet. Scale bar 1 cm. Source data are provided as a Source Data file.

## Scope of the photothermal approach

After developing a working procedure for the photothermal synthesis of spherical IONPs, we aimed to compare our methodology with conventional protocols and expand its scope to different morphologies (Fig. 3). Our results and proposed mechanism suggest that the light-assisted synthesis could yield nanoparticles at overall lower temperatures. To test this, we set up eight identical solutions as described before (Fig. 1, Methods), four of which were heated by light treatment and the remaining four by a hotplate. The results, summarized in Fig. 3a, show a clear trend indicating that the photothermal synthesis requires lower temperatures to achieve nucleation and produce similar-sized nanoparticles (Supplementary Fig. 29). Following these encouraging results, we attempted to extend the comparison to additional conditions and morphologies. For that, we modified the synthesis described in Fig. 1a to yield triangular iron oxide nanoparticles (TIONPs) by adding sodium oleate, a known growth-directing reagent (see Methods)[42]. The photothermal synthesis yielded ≈6–8 nm TIONPs at 180 °C while the conventional synthesis required a temperature of ≈210 °C to attain similar particles (Fig. 3b, c, f, and Supplementary Fig. 30). Importantly, no significant difference in monodispersity and non-triangular impurities could be identified between the photothermal and conventional syntheses. Intriguingly, the photothermal approach consistently produced a higher

nanoparticle yield when compared to the conventional synthesis (Supplementary Fig. 31a, b).

The notable ability to lower the overall nucleation temperature and increase reaction yields demonstrated by the plasmonic photothermal methodology (Supplementary Fig. 31a, b), prompted us to test whether we could decrease the reaction temperature of known protocols. Hence, we chose two reaction procedures that showed IONP formation at a minimal temperature of ≈300 °C[39,42]. For the first reaction, iron stearate was used as the precursor for small spherical nanoparticles, in the second reaction, the decomposition of iron oleate resulted in hexagonal plate structures (see Methods). Intriguingly, both light-assisted reactions proceeded at 200 °C, representing a ≈100 °C difference from the conventional syntheses, without noting any significant influence on morphology (Fig. 3b, d, e). The photothermal reaction in the presence of iron stearate led to slightly smaller spherical nanoparticles of size ≈2–4 nm in comparison to conventional heating (≈3–4 nm) (Fig. 3d, g). In contrast, the nanoplate-like morphologies were of an average size of ≈15 ± 3 nm in both types of syntheses (Fig. 3e, h). It is worth mentioning that both reactions were also carried out in the presence of SiO₂@AuBPs by conventional heating at 200 °C and did not afford any noticeable nanoparticle formation. These results highlight the ability of the photothermal method to considerably decrease the activation energy necessary for

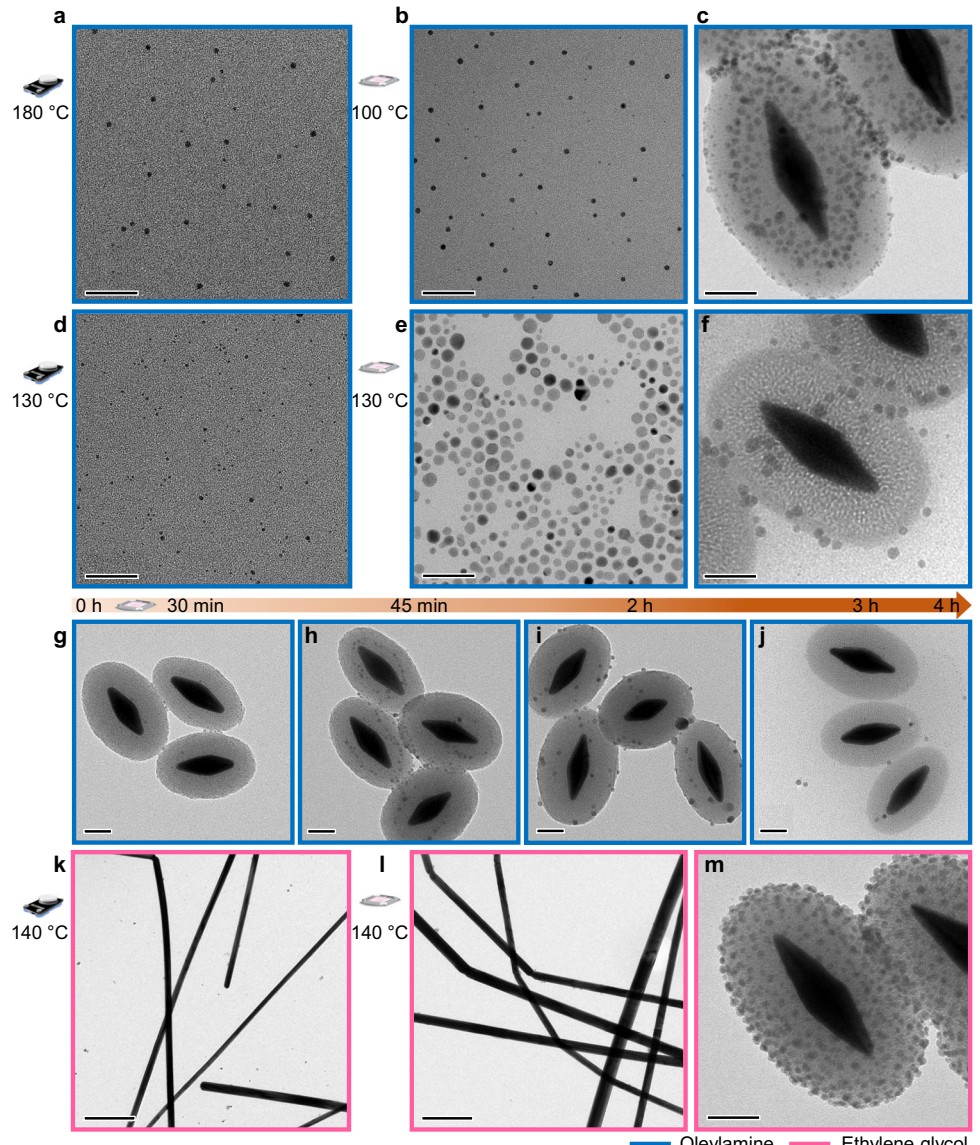

**Fig. 4 | Photothermal synthesis of silver nanoparticles/wires. a–f** TEM images of silver nanoparticles synthesized in oleylamine (blue box). **a, d** using conventional heating, and (**b, c, e, f**), under photothermal conditions (**b, e**), with corresponding silica-encapsulated AuBPs decorated with the particles (**c, f**). Scale bars 50 nm. **g–j** Time-dependent (indicated by the brown arrow) TEM analysis of the AgNP formation (at the photothermal temperature of 130 °C) over the AuBP surface showing a gradual decrease in nanoparticle concentration on the surface: **g**, 30 min; **h**, 45 min; **i**, 2 h; and **j**, 3 h. Scale bars 50 nm. **k** TEM images of silver nanowires synthesized in ethylene glycol solvent (red box) under conventional heating with a small number of nanoparticles as a side product. **l–m** Silver nanowires synthesized under photothermal condition (**l**), with corresponding silica-encapsulated AuBPs **m**, decorated with silver nanoparticles produced as a byproduct. Scale bars for (**k, l**) 0.5 μm, and (**m**) 50 nm.

nanoparticle formation without impairing the selective growth mechanism leading to diverse structures with high yields (Supplementary Fig. 31c).

### Generalizing light-assisted nanoparticle synthesis to metals

The photothermal synthesis established for IONPs, displayed both high efficiency and the potential to access hybrid materials. Consequently, we strived to generalize our approach to silver and palladium nanoparticles (AgNPs and PdNPs). For silver, we modified two syntheses from known procedures to afford spherical and wire-like nanoparticles (AgNSs and AgNWs, see Methods, and Fig. 4)[45,46]. The former was initially carried out in oleylamine at 180 °C with a conventional set-up, yielding ≈3–5 nm nanospheres (Fig. 4a). Notably, under light irradiation AgNSs of similar size were formed at 100 °C (Fig. 4b, and Supplementary Fig. 32). Increasing the photothermal temperature to

130 °C promoted additional growth to ≈4–12 nm, while decreasing the conventional synthesis to the same temperature resulted in ≈2–4 nm particles (Fig. 4d, e, and Supplementary Figs. 32, 33). These results correspond with our previous observations on IONPs; light-assisted particle formation proceeds at lower bulk temperature and enables increased growth at comparable conditions. Moreover, hybrid structures, resembling the previously described IONP@AuBP, could be isolated (Fig. 4c, f, and Supplementary Figs. 34–36) and careful analysis of their TEM images revealed they underwent similar processes. AgNSs attached to their surface were larger (≈5–7 nm at 100 °C, ≈8–12 at 130 °C) than the particles in the solution, and increasing the photothermal reaction temperature led to less AgNPs on the silica surface (Fig. 4c, f). To corroborate the suggestion that indeed AgNPs are subjected to processes analogous to iron, we conducted a time-dependent evaluation of the silver-decorated hybrid particles,

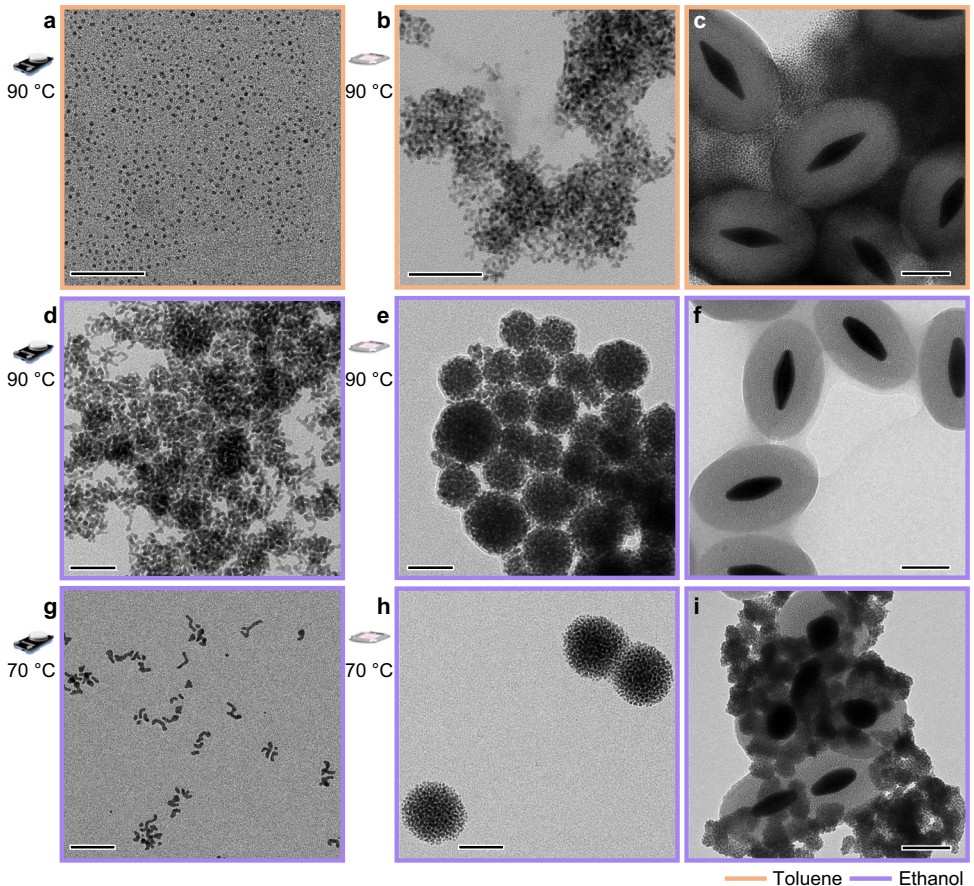

Toluene ▬ Ethanol

**Fig. 5 | Photothermal synthesis of palladium self-assemblies. a** TEM image of palladium nanoparticles synthesized in toluene (orange outline indicating solvent) at 90 °C under conventional heating. **b** TEM image of elongated palladium nanoparticles synthesized via light treatment at 90 °C. **c** Silica-encapsulated AuBP nanoparticles isolated from the photothermal reaction in toluene. **d** Aggregated palladium nanorods synthesized in ethanol (violet outline) under conventional heating at 90 °C. **e** Palladium self-assemblies obtained from the photothermal reaction at 90 °C. **f** Isolated SiO$_2$@AuBPs from the same reaction. **g** Palladium nanorods synthesized in ethanol at 70 °C under conventional heating. **h** Self-assemblies produced in the photothermal reaction at 70 °C. **i** Isolated SiO$_2$@AuBPs from the same reaction. Scale bars for (**a**, **b**, **d**, **e**, **g**, **h**) 50 nm, and (**c**, **f**, **i**) 100 nm.

showing again that when the reaction matures the hybrid structure dissociates (Fig. 4g–j). Thus, we determined that the mechanism proposed in the IONP case was also valid for the formation of AgNPs. In order to establish further scope, we found anisotropic AgNWs could also be synthesized with light, resulting in slightly thicker (≈20–30 nm on average) nanowires when compared to conventional synthesis (Fig. 4k, l, and Supplementary Figs. 37–38). However, during the growth of the nanowires they were not found to be attached to the AuBPs possibly due to steric complexity. Interestingly, in the photothermal synthesis of nanowires the AgNS byproduct (≈4–6 nm) homogeneously assembled on the SiO$_2$@AuBPs and could hardly be observed separately (Fig. 4m), whereas in the conventional case, the byproduct was distributed throughout the imaged TEM grid (Fig. 4k). When the nanowires were conventionally synthesized in the presence of SiO$_2$@AuBPs, the spherical impurities did not appear to be bound to the plasmonic nanoparticles, emphasizing again the distinctive effect of light irradiation. It is also worth mentioning that we were able to obtain higher yields of nanoparticles under photothermal conditions at a much lower overall temperature compared to conventional synthesis methods as previously observed for IONPs (Supplementary Fig. 39).

Subsequently, we directed our efforts to investigate the photothermal synthesis of PdNPs, which generally require lower temperatures than those utilized previously. Thus, PdNPs were synthesized using conventional heating in toluene and ethanol with palladium acetate as a precursor in the presence of the stabilizing agent

n-dodecyl sulfide (Fig. 5, see Methods)[47]. Conventional heating at 90–95 °C in toluene resulted in ≈3–4 nm nanoparticles as reported in the literature (Fig. 5a)[47]. Intriguingly, under identical photothermal conditions slightly elongated, clustered nanoparticles were obtained (Fig. 5b, and Supplementary Fig. 40). TEM images showed the disordered clusters also formed around the SiO$_2$@AuBPs utilized in the reaction (Fig. 5c, and Supplementary Fig. 41). When ethanol served as the solvent, conventional heating conditions yielded Pd rod-like structures (≈15–20 × 3–5 nm) at 70 and 90 °C with the latter showing clusters similar to the photothermal reaction in toluene (Fig. 5d, g). Remarkably, the photothermal reaction in ethanol at both temperatures led to the self-assembly of PdNPs into superstructures with a well-defined spherical shape of ≈45–65 nm (Fig. 5e, h, and Supplementary Figs. 42, 43). The SiO$_2$@AuBP samples isolated differed from each other, the silica surface at the lower temperature was covered with the described palladium assemblies, whereas at 90 °C it remained free of any particles, agreeing with our previous observations (Fig. 5f, i, and Supplementary Figs. 44–46). The aggregating effect observed in the toluene via the light-assisted reaction may be due to the SiO$_2$@AuBP serving as nucleation centers, increasing the local concentration of both precursor and reduced particles around the AuBPs, facilitating the occurrence of clusters. In ethanol, where the solvent can also act as a reducing agent, an elevated temperature seemed to be the contributing factor for the agglomeration observed under conventional conditions. This did not recur when light was used to heat the reaction, possibly the combination of a reducing solvent and

localized elevated temperatures around the AuBPs creates a pocket with an increased reduction rate where Pd-seeds form in a confined space leading to the self-assembly[48,49]. When conducting control experiments to probe the reaction's behavior when treated with light without AuBPs, we noticed that in an ethanolic environment nanoparticle formation occurred under ambient temperature. However, the product of reactions with and without light treatment in the absence of AuBPs was different. When the reaction was irradiated, a sudden rise in the temperature was observed due to the formation of PdNP seeds, which now acted as photothermal agents. This led to clustered nanoparticle formation partially resembling the product obtained from AuBP-initiated reactions. In contrast, when the reaction was left at room temperature no clustering could be seen and a more dispersed pattern was observed in TEM images (Supplementary Fig. 47). The unexpected outcome of the routine control highlights the crucial role of photothermal heating and subsequent nucleation in the close proximity have in forming clustered assemblies. The formation of palladium self-assemblies upon utilizing the photothermal approach with high yield (Supplementary Fig. 48), obviating the need for complicated directing strategies, demonstrates that the light-assisted reaction follows an alternative pathway with an immense potential to open innovative research opportunities. The presented research can be further applied to practically numerous kinds of colloidal nanoparticles with diverse synthesis conditions (Supplementary Fig. 49) affording insights limited previously through conventional heating methods.

In our work, we established the use of plasmonic photothermal activation of $SiO_2$@AuBPs as a practical option for the synthesis of colloidal nanoparticles with low-energy NIR LEDs as the light sources. An in-depth analysis indicated that the photothermal methodology proceeds through a distinctive mechanism enabling two major advantages. First, the light-assisted synthesis afforded nanoparticle formation at lower temperatures by decreasing the nucleation energy and permitting faster growth kinetics, improving the overall efficiency of the process. Second, applying the light-assisted approach allowed access to diverse superstructures beyond the reach of parallel conventional syntheses. We intend to continue investigating the processes governing the plasmonic photothermal method and utilize this knowledge to rationally design the synthesis of materials for different applications.

## Methods

### List of materials
All materials were purchased from Sigma-Aldrich unless noted otherwise, and used as it is without further purification. Ultrapure water (type 1, 18.2 MΩ) from Millipore® Direct-Q® 3 with UV was used.

Cetyltrimethylammonium chloride (CTAC), sodium borohydride (ReagentPlus, 99%), sodium citrate tribasic (BioUltra, ≥99.5%), gold chloride trihydrate (99.9%), cetyltrimethylammonium bromide (CTAB, ≥99%), ascorbic acid (BioXtra, ≥99.0%), hydrochloric acid 32%, silver nitrate (BioXtra, ≥99%), tetraethyl orthosilicate (TEOS, reagent grade 98%), ammonium hydroxide (28% in water 99.9%), ethanol (EtOH, 99.9%, tech Romical), iron (III) chloride hexahydrate ($FeCl_3 \cdot 6H_2O$, ACS reagent 97%) were used.

### Preparation of silica-coated AuBPs
Seed solution and gold bipyramids were synthesized via seed-mediated growth method described by Sánchez-Iglesias et al.[12] 5 mL of $HAuCl_4$ (10 mM), 1 mL of $AgNO_3$ (10 mM), and 2 mL of HCl (1 M) were added to a 100 mL solution of CTAB (100 mM) in water. The pre-made seed solution was added right after 0.8 mL of L-ascorbic acid was added with vigorous stirring. Control over AuBP size was achieved by varying the seed solution concentration. The reaction was set at 30 °C for 2 h. Then, the AuBP solution was centrifuged 3 times at 7000 × g for 15 min with 1 mM CTAB in water. The encapsulation of fresh

synthesized AuBPs included two processes: First, the mesoporous silica encapsulation was made by adding 30 μL of 0.1 M NaOH and 10 μL of TEOS, to a 10 mL solution of 0.3 mM CTAB and 2 OD AuBPs (measured at maximum LSPR), then, the solution was shaken overnight at 120 rpm and 30 °C. After that, the mesoporous encapsulated AuBPs were centrifuged 4 times with EtOH at 7000 × g for 10 min. The second process, Stöber encapsulation, was made by adding to a 10 mL solution of 4 OD AuBPs (measured at maximum LSPR) in EtOH 250 μL 28% ammonia in water and 25 μL of 20% TEOS in EtOH were added and the solution was set in a shaker at 120 rpm and 30 °C for 12–24 h. Lastly, the solution was washed 3 times with EtOH and concentrated for use. Considering OD is a substitute for concentration then OD volume is substitute for amounts, and considering OD-times volume in the synthesis steps as 100%, the yield of the final particles is ≈30%. A correlation between OD and the actual concentration of gold is shown in Supplementary Fig. 50, specifically, for 1 OD (measured at 850 nm) solution of $SiO_2$@$AuBP_{850}$, the concentration of Au is ≈19.2 ± 0.1 ppm which corresponds to ≈0.1 mM of Au and approximately 10 pM of AuBPs.

### Synthesis of silica-coated $AuBP_{660}$ and $AuNR_{850}$
Gold bipyramids of LSPR ≈ 660 nm ($AuBP_{660}$) were synthesized following the same procedures as described for $AuBP_{850}$ simply by varying seed solution concentration in the growth solution. Gold nanorods of LSPR ≈ 850 nm ($AuNR_{850}$) were synthesized according to the procedure reported in the literature[50], with slight modification. Briefly, 1.6 mL of $HAuCl_4$ (6 mM) was added to 8 mL of aqueous CTAB solution (200 mM). Next, 160 μL of $AgNO_3$ (10 mM) and 120 μL of ascorbic acid (100 mM) were added stepwise followed by the addition of 40 μL of $NaBH_4$ (1 mM) solution. The mixture was kept until the color of the solution changed to deep brown. The AuNRs were further purified by centrifuging at 7000 × g for 5 min and washed with water. Finally, both the $AuBP_{660}$ and the $AuNR_{850}$ were coated with silica similar to the $AuBP_{850}$. The silica coating assists in dispersibility in the different reaction medium, improves heat dissipation into the medium surrounding the plasmonic nanoparticle surface, and shield them from major degradation (Supplementary Figs. 1, 2, 10).

### Synthesis of silica NPs
Silica NPs were synthesized following the Stöber method with slight modification[51], specifically, 5 mL of MeOH, 15 mL of 2-propanol, and 4.2 mL of 25% ammonia solution were mixed under magnetic stirring and the reaction was continued for 30 min. Next, the NPs were centrifuged at 3000 × g for 2 min and washed extensively with MeOH before imaging by electron microscopy (Supplementary Fig. 2).

### Preparation of iron-oleate/stearate precursor
Fatty acid metal salts mainly iron oleate/stearate were prepared by reacting $FeCl_3$ $6H_2O$ with oleic acid/stearic acid respectively in the presence of a methanolic solution of tetramethyl ammonium hydroxide (TMAH). Briefly, 4 mL of stearic/oleic acid was dissolved in MeOH and mixed with 6.3 mL of TMAH (25 wt.% in MeOH). Next, the solution was heated to 50–60 °C, and subsequently, 0.8–1.3 g of iron salts dissolved in MeOH was added dropwise. The brownish precipitate obtained was washed several times with MeOH and dried in a vacuum to get a waxy powder of iron oleate/stearate.

### Photothermal system set up
All LEDs were purchased from LCFOCUS and a 9130B-BK precision programmable DC power supply was used to connect the LEDs to a computer. The temperature was monitored by MLX90614-DF ROBOT IR thermometer sensor, which was connected to a computer via an Arduino Mega (Supplementary Fig. 3). The different parts of the system were controlled with a custom LabVIEW program.

## Photothermal synthesis of iron oxide NPs

Colloidal iron oxide NPs were synthesized using the photothermal heat generated by the plasmonic AuBPs. 100-150 mg of iron oleate were dissolved in 0.5 mL octadecene (ODE) and loaded in a 20 mL glass vial/flask, and purged with argon for 10 min. Then 2.5 mL oleic acid-oleylamine (60% with reference to amine) mixture was added, and the solution was further degassed at 80 °C for 30 min. The solution was cooled naturally at room temperature and magnetically stirred in the presence of AuBPs (15–20 OD, measured at 850 nm) under continuous illumination of light (850 nm) at inert condition. The photothermal heat/temperature at 160–200 °C was maintained for 1–4 h by simply controlling the light illumination time using a thermometer sensor (see above). Finally, the light illumination was stopped, and the colloidal nanoparticles were extracted with chloroform/toluene, centrifuged at 3000 × $g$ for 5 min to separate from the AuBPs, and the supernatant was collected, and subsequently precipitated with EtOH. The isolated precipitate was separated using a laboratory-based bar magnet and dissolved again in 0.5 mL toluene followed by further precipitation using 1 mL of EtOH. This step was repeated twice, and finally, the purified NP solution was prepared in toluene and used as a stock solution for further use. The precipitate containing AuBPs was further washed thoroughly with chloroform/toluene twice to remove free/loosely bound particles and dispersed in EtOH under mild sonication for further use.

## Synthesis of octadecyltrimethoxysilane (OTMS) coated AuBPs

OTMS functionalized AuBPs were obtained by our reported approach with slight modification[44]. Briefly, reacting a methanolic solution of AuBP (18–20 OD, measured at 850 nm) with 150–180 μL of OTMS in 1 mL toluene, followed by heating at 60–70 °C. 15-18 μL of tetramethyl ammonium hydroxide (TMAH) was mixed with 1 mL of MeOH and added dropwise to the above solution. The reaction was continued for ≈30 min under stirring condition until all the AuBPs were precipitated. The isolated precipitate was washed 2–3 times with toluene and EtOH alternatively, and finally can be used to disperse in a wide range of nonpolar solvents for further use.

## Shape-controlled photothermal synthesis of iron oxide NPs

For synthesizing iron oxide nanotriangles, a similar synthesis procedure was followed as described for iron oxide NPs only in the presence of sodium oleate. Briefly, 150 mg of iron oleate, and 50 mg sodium oleate were dissolved in 0.5 mL ODE and loaded in a 20 mL glass vial/flask, followed by purging with Ar for 10 min. Then 2.5 mL oleic acid-oleylamine (60% with reference to amine) mixture was added, and the solution was further degassed at 80 °C for 30 min. The solution was cooled naturally at room temperature and magnetically stirred in the presence of AuBPs (15-20 OD, measured at 850 nm) under continuous illumination of light (850 nm) at inert condition. The photothermal heat/temperature at 180 °C was maintained by controlling the light illumination time using a thermometer sensor (see above). The NPs and AuBPs were purified by EtOH/acetone-based precipitation and toluene/chloroform-based redispersion as described for iron oxide NPs.

Iron oxide NPs using iron stearate as precursor were also synthesized from the reported literature[39] with slight modifications. Briefly, 373 mg iron stearate, 161 mg octadecylamine, 161 mg N-methylmorpholine N-oxide were mixed with 3 mL of octadecene, degassed by purging with Ar for 15–30 min in vacuum, followed by mixing with 15-20 OD AuBP (measured at 850 nm) and magnetically stirred in under continuous illumination of light (850 nm) at inert condition. The photothermal heat was maintained and regulated at 200 °C for 30 min. The NPs were purified by EtOH/acetone-based precipitation and toluene/chloroform-based redispersion three times.

Iron oxide hexagonal nanosheet-like structures were synthesized from the reported literature[42] with some modifications. Briefly, 300 mg iron oleate and 60 mg sodium oleate were mixed with 54 μL of oleic acid and 5 mL of ODE followed by degassing at 80 °C for 30 min under a continuous flow of Ar. The solution was cooled naturally at room temperature and magnetically stirred in the presence of AuBPs (15-20 OD, measured at 850 nm) under continuous illumination of light (850 nm) at inert condition. The photothermal heat was maintained and regulated at 200 °C for 2.5 h. The NPs were purified by EtOH/acetone-based precipitation and toluene/chloroform-based redispersion thrice.

## Photothermal synthesis of silver NPs/wires

47 mg of silver nitrate was dissolved in 10 mL oleylamine, and 50 μL of oleic acid was added[45]. The reaction was degassed at 50 °C by purging with Ar, and subsequently cooled to room temperature, and further mixed with AuBP (15 OD, measured at 850 nm) followed by illumination with a LED of 850 nm at inert condition. The photothermal temperature was maintained and regulated at 100-180 °C for 1 h. The NPs were purified by extracting them in chloroform/toluene, centrifuged at 3000 × $g$ for 5 min, and the supernatant was precipitated using EtOH. The process was repeated twice, and finally, the purified NPs can be stored in chloroform/toluene.

Silver nanowires were synthesized by the reduction of silver nitrate in the presence of polyvinylpyrrolidone (PVP) in ethylene glycol[46]. 50 mg PVP of molecular weight ($M_w$) 55,000 g mol$^{-1}$, and 100 mg PVP of $M_w$ 360,000 g mol$^{-1}$ were dissolved with 25 mL of ethylene glycol. Subsequently, 250 μL of FeCl$_3$ solution (600 μM in EG) and 180 mg of AgNO$_3$ were rapidly added in the above solution within 1 min. The reaction mixture was further mixed with AuBP (15 OD, measured at 850 nm) and treated with the light source, maintaining a temperature of 140 °C for 50 min. The nanowires were precipitated by centrifugation and washed with EtOH.

## Photothermal synthesis of palladium NPs/assemblies

Palladium NPs were synthesized using palladium acetate in the presence of n-dodecyl sulfide following the reported literature[47] with slight modifications. 1.5–5 mg of palladium acetate and 3.5–14 mg of n-dodecyl sulfide were mixed with 3-5 mL of toluene or EtOH in the presence of AuBP (2–4 OD, measured at 850 nm). The solution was heated at 70–90 °C under continuous illumination of light using an 850 nm LED in a closed vial. The photothermal temperature was maintained for 3 h and the product was purified by centrifugation followed by washing with toluene/EtOH.

Palladium nanocubes/bars were synthesized in aqueous solution according to the procedure reported in the literature[52], with slight modification (Supplementary Fig. 49d–f). Briefly, 8 mL of an aqueous solution containing 105 mg PVP of molecular weight ($M_w$) 55,000 g mol$^{-1}$, 60 mg L-ascorbic acid, and 300 mg KBr were placed in a 20 mL vial, and mixed with 3 mL of an aqueous solution containing 57 mg of sodium tetrachloropalladate (II) (Na$_2$PdCl$_4$). The solution was further mixed with AuBPs (2 OD, measured at 850 nm) and treated with the light source (850 nm), maintaining a temperature of 60 °C for 3 h in a closed vial. The product was purified by centrifugation followed by washing with water.

## Instruments

**UV–Vis spectrophotometer.** Thermos-Scientific Evolution 220 UV–Visible spectrophotometer was used to determine AuBPs solutions' optical properties and optical density (OD). Samples were diluted to obtain accurate measurements and solvents varied depending on experiment requirements. UV–Vis absorbance spectra of the iron oxide NPs (Fig. 1h) exhibit absorbance mainly in the UV region (300–400 nm) due to ligand (oxygen species) to metal (Fe (III)) charge transfer, and a weak visible absorption (>400 nm) due to $d$–$d$ transition within Fe (III) in the iron oxide NPs.

**FTIR spectrometer.** Fourier-transform infrared spectroscopy (FTIR) measurements were performed on a Thermo Scientific Nicolet 6700 spectrometer in ATR mode.

**Zeta potential.** A Zetasizer (NanoZS, Malvern) was employed to measure the surface charge of NPs.

**Transmission electron microscope (TEM).** A Tecnai T12 G2 TWIN TEM Thermo Fisher Scientific (former FEI) Transmission Electron Microscope was mostly used for general imaging purposes using Gatan CCD MultiScan camera. Thermo Fisher Scientific (FEI) Talos F200C transmission electron microscope operating at 200 kV was used for both imaging and tomography data accusation. The images were taken with Ceta 16 M CMOS camera. The tilt series were acquired with Thermo Fisher Scientific Tomography software (version 5.9) between angles varying from −50° to 50° with 1° steps. Before each acquisition of an image and before the movement of the stage to the next tilt angle, a cross-correlation of tracking and focusing were made. The post-processing and the stack alignment were done with Thermo Fisher Scientific Inspect3D software (version 4.4). The JEM-2100F, a Field Emission gun Transmission Electron Microscope was employed to measure Energy-dispersive X-ray spectroscopy (EDS) elemental mapping under scanning transmission electron microscope (STEM) mode using Oxford EDS system, and obtain high-resolution images. The post-processing of the images in selected cases was performed using ImageJ software. TEM grids were prepared by adding 3–8 μL of purified liquid samples in different solvents on Electron Microscopy Sciences ultrathin/formvar/carbon 200 Mesh, copper grids and letting the liquid evaporate under air.

**X-ray diffraction (XRD).** Panalytical Empyrean II Diffractometer system equipped with three position sensitive detectors: X'celerator 1D, 1der (0D and 1D applications), and PIXcel3D detector (with premounted diffracted beam monochromator) was employed to measure XRD data. For the XRD measurements, samples were prepared by drop-casting a film of the nanocrystals on glass slides from the solution of nanocrystals dispersed in hexane/chloroform. The XRD patterns in Fig. 1e demonstrate a broad feature inherent due to nanostructured materials with peak positions at $2\theta = \approx 30.3, 35.6, 43.3, 53.5, 57.3,$ and 62.8 corresponding to the (220), (311), (400), (422), (511), and (440) planes of γ-$Fe_2O_3$[39].

**Inductively coupled plasma optical emission spectrometry (ICP-OES).** The amounts of Au and Fe in IONP@AuBP samples were determined using a SPECTRO ARCOS ICP-OES. The samples were prepared by dissolving the samples in 2 mL of aqua regia and then diluting to obtain a 10% nitric acid solution. Similarly, the relationship between the OD (measured at 850 nm) and Au concentration of AuBP₈₅₀ was determined by analyzing samples with different known ODs with the ICP-OES. Synthetic yields of the nanostructures were determined with the purified products using ICP-OES-based analysis with detection limits of 0.01 mg kg⁻¹ (10 ppb). The products were digested in aqua regia followed by diluting to obtain 10% nitric acid solutions, and the dilution factors ($f_{dil}$) were carefully noted for each analysis. Nanoparticle yield was obtained from the concentration ($C$) of the corresponding elements obtained from the ICP analysis using the following equation,

$$\frac{C_M \text{ in product}}{C_M \text{ in precursor}} \cdot f_{dil} \cdot 100 = \%\text{yield}, M = \text{Fe, Ag, Pd} \quad (1)$$

**X-ray photoelectron spectroscopy (XPS).** ESCALAB Xi+ Thermo-Fisher Scientific ultrahigh vacuum ($1 \times 10^{-9}$ bar) with Al Kα X-ray source was used to investigate the surface chemical composition, and high-resolution core-level spectra were utilized to probe the

corresponding valence states of each component element. All the spectra were charge corrected by using the C 1s line at 284.6 eV, which appears due to the presence of carbon on the sample surface. The XPS spectra of the samples were further analyzed using the Thermo Scientific AVANTAGE software. X-ray photoelectron spectroscopy (XPS) of the nanoparticles in Fig. 1f–g represents core-level scanning of Fe 2p (Fig. 1f) and O 1s (Fig. 1g); Fe 2p spectra exhibit two peaks, at ≈724.5 eV for Fe $2p_{1/2}$ and ≈710.9 eV for Fe $2p_{3/2}$, which are the characteristic peaks of the $Fe^{3+}$ ion of γ-$Fe_2O_3$[40]. Moreover, the absence of any signal or shoulder at smaller binding energies, as would be expected for the presence of the $Fe^{2+}$ ion (≈708 eV), confirms the absence of any impurity phase related to $Fe^{2+}$ in the nanocrystals. This can be further verified by the additional peak of ≈718.7 eV as the shakeup satellite peak, which is absent for the $Fe^{2+}$ ion. The lattice and surface oxygen atom due to the capping ligand were verified from the signals at ≈530.1 eV, and ≈531.6 eV respectively[41]. The atomic ratio of iron and lattice oxygen in the nanoparticle was found to be 2:3.2.

## Data availability
The data that support the findings of this study are available from the corresponding author upon request. Source data are provided with this paper.

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

## Acknowledgements

The authors thank Dr. Vladimir Ezersky for assistance with JEM-2100F Transmission Electron Microscope. We thank the Ilse Katz Institute for Nanoscale Science and Technology for the other technical support of material characterization. Y.W. acknowledges the support of the Zuckerman STEM Leadership Program, and the Israel Science Foundation (ISF), grant No. 2491/20. Figure 1 (agreement number: UQ25X168GR), and Fig. 2 (agreement number: IZ25X168S1) were created with BioRender.com.

## Author contributions

Y.W. supervised the project. A.B. conducted all the experiments. E.Y., N.L., and O.S. set up the LED temperature regulation system. N.L., A.B., O.S., D.Y., and E.Y.F. synthesized the AuBP nanoparticles.

A.U. performed the TEM tomography. A.B. took the TEM images. A.B., N.L., and Y.W. wrote the manuscript.

## Competing interests

The authors declare no competing interests.
