## [Peer Review File · Nature Communications]

nature portfolio

Peer Review FileReviewer comments, first round -

Reviewer #1 (Remarks to the Author):

The work "Unlocking the power of plasmonic photothermal activation to custom colloidal synthesis" by Biswas et al. shows an interesting wet chemistry synthesis protocol to obtain nanoparticles that typically are obtained by heating up the reaction solution, but instead using the photothermal properties of gold nanoparticles, in this case using silica-coated nano-bipyramids. With this protocol, they synthesized iron oxide, silver, and palladium nanoparticles. This approach is not new, as plasmonic nanoparticles have already been previously used to synthesize other nanoparticles or to grow them with initial seeds with the same or other materials (see for instance, ACS Nano 2018, 12, 8447–8455, J. Am. Chem. Soc. 2017, 139, 19, 6771–6776, J. Phys. Chem. C 2018, 122, 50, 28901–28909, Nature Materials (2016) 15, 889–895). In this work, the authors synthesize different types of nanoparticles and claim that with this approach they can decrease the temperature of the reaction with respect to the standard synthesis. The system is not very practical, as it requires the previous synthesis of the complex nanoheaters (which require several steps and some experience in the synthesis of nanoparticles), and lacks the simplicity of just using a hotplate. I am saying this because, they are using only the photothermal effect, but not other interesting properties of the plasmonic-driven reactions like the generation of hot electrons, or electromagnetic hotspots (the silica surrounding the gold core, most probably prevents it). Despite this, the approach used here could still be interesting as they obtained other types of nanoparticles, such as the core-satellite structures (Au@SiO₂-FeOx) that could be of interest for multifunctionality in different applications.

Regarding the manuscript I have not found any clear flaw in it, but I have a few questions/comments.

1. In the synthesis of the nanoparticles, they did a double growth of the silica shell, first as porous and then as a continuous shell. Why is that? What is the function of the mesoporous inner shell?
2. The quantities of nanoparticles used for the synthesis are given in Optical Densities. What is the wavelength used to calculate this OD? The OD, could be useful to measure the photothermal capacity of the reaction solution, however, it would be useful (from the reproducibility point of view) to know the concentration in terms of Au moles and in terms of moles of nanoparticles (at least an estimation).
3. In line with the previous comment. The authors say on page 3, that the temperature of the reaction could be changed by changing the power or the concentration of nanoparticles. I guess that as the heat is higher at the surface of the nanoparticles, the more nanoparticles, the lower their temperature should be (as there is more surface available), so the results will depend on the nanoparticle's concentration. Have the authors tested any of the syntheses varying the nanoparticle concentration?
4. As the photothermal effect at the nanoparticle surface seems to be the key for the synthesis at lower temperatures, could the authors estimate what temperature that would be? (I mean in comparison with solution temperature).
5. The synthesis of iron oxide at the silica surface and then detachment seems quite interesting, however, I have difficulties imaging this happening for the Ag nanowires, how does this happen? From what is appreciated in the TEM images it looks like both nanowires and nanospheres are obtained in the reaction, and then the nanospheres end up sticking to the silica.

Reviewers #2 and #3, who co-reviewed the work (Remarks to the Author):

The authors describe a methodology to synthesize different colloidal nanoparticles, such as iron oxide, silver, and palladium, using gold bipyramids as plasmonic heaters capable of inducing nanoparticle formation via a photothermal process. One of the key aspects is the stabilization of gold bipyramids with a silica shell to stabilize them in different organic solvents where the nanoparticle growth occurs. Notably, the authors demonstrate that photothermal temperatures between 50 and 100 °C lower than those using estándar heating approaches are required for the synthesis processes. Furthermore, a mechanism based on the nucleation of nanoparticles on the

silica surface and detachment during the late stages of the synthesis is proposed in the case of the iron oxide system.

Although the concept of plasmonically mediated heating for the synthesis of colloidal nanomaterials is interesting, I think the quality of the obtained nanoparticles (iron oxide, silver, and palladium) is not sufficiently high compared with many current protocols found in the literature. In addition, important control experiments are missing, and some questions remain open. For instance, what would happen if the synthesis of nanoparticles is attempted in the absence of gold bipyramids but in the presence of silica nanoparticles of similar dimensions? The plasmon band of gold bipyramids is located at ca. 850 nm before irradiation experiments, resonating with the LED light source. However, after 45 min, the bipyramid reshaping shifts the band to 650 nm. Since the synthesis of iron oxide nanoparticles requires at least 4 h, how is the heating of the solution maintained when the plasmonic band is, after 45 min, off-resonance with the light source?

A 100 W light source focused on a small vial, such as those used for the synthesis, implies a high amount of energy on a small volume. Is it possible that the solvent or reactant interaction with such a light source promotes the nucleation and growth of iron oxide (silver and palladium also) nanoparticles?

Could it be that free radicals are generated during the irradiation process (in the presence and absence of gold bipyramids) and participate in nanoparticle growth?

In my view, the scalability of the proposed synthesis methods is limited by the need to concentrate a powerful light source on a small volume of growth solution. Moreover, the amount of gold bipyramids is quite high. An OD of 15-20 implies a concentration of Au between 2.5 and 3.5 mM, which is quite demanding. Do the authors have a potential solution to these issues?

More importantly, the discussion includes the concept of "catalysis". However, it should be clarified why the described photothermal synthesis can be considered as a catalytic process and not simply as a heterogeneous nucleation process, for instance there is not deep explanation about the reactivity mechanisms involved in the process.

For these reasons, I would not recommend publication in Nature Communication but in a sister journal such as Scientific Reports, after the issues mentioned above are addressed.

Reviewer #4 (Remarks to the Author):

The paper reports the use of plasmonic Au nanopyramids coated with SiO₂ as a heater to heat a reaction solution of nanoparticle precursors to form the desired nanoparticles. Examples on light-induced synthesis of iron oxide, silver, and palladium nanoparticles were given. The photothermal method was also extended to synthesize Ag nanowires, suggesting its potential to achieve anisotropic growth of nanostructures. The reported photon driven nanoparticle synthesis is of some interest, but the synthesis outcome is not very competitive to the well-established thermal synthesis method. The work does not show high enough novelty or importance for this journal.

It is not clear why a special feature of nanostructured Au, Au bipyramids (AuBPs), was selected as a plasmonic heater. What is unique part of these bipyramids over Au nanorods? The justification given by the authors "as they exhibit tunable narrow localized surface plasmon resonance (LSPR) bands and can be synthesized with high yields.¹⁰⁻¹²" does not seem to justify the uniqueness and importance of this selection as Au nanorods with similar optical properties have been prepared in large scale.

The main novelty claimed by the authors is that using the plasmonic heating can lower the reaction temperature for nanoparticle synthesis. This is certainly interesting, but the degree of the temperature drop demonstrated in this paper is not that significant. This, plus the complexity introduced by Au nanopyramids synthesis, SiO₂ coating, and light irradiation, does not provide much advantage over the conventional thermal heating approach to nanoparticle synthesis.

In the synthetic section, there is no report on synthetic yields on nanostructures made by their light-induced synthesis.

In the section "Catalytic behavior, and scope of the photothermal approach", what catalytic behavior the authors are referring to?

In the synthesis of Ag nanoparticles, why does the 250 μ L of FeCl₃ solution need to mix with 180 mg of AgNO₃ in order to prepare Au nanoparticles? What is the role of FeCl₃? Is there any Fe involvement in the final Ag nanostructures?

Reviewer #5 (Remarks to the Author):

Authors present an original method for monodisperse nanoparticles synthesis. Using a photothermal approach involving plasmonic effect of silica coated gold bipyramidal nanoparticles (AuBP), they successfully managed to synthesise iron oxide, silver and palladium NPs with different sizes and shapes. Besides, they studied the effect of different parameters of the protocol (temperature, exposure time, concentration of reagents) and showed that their approach requires less energy than standard approaches using a hotplate.

Although I am supportive, I do have some critics:

1 Authors claim that they could synthesise NPs at lower temperature than standard approach. As an IR thermometer has been used, the temperature measured is rather an average temperature depending on resolution of the camera (which should be few μ m). That is the reason why, I doubt that the temperature at the vicinity of the AuBP NPs is such lower. Indeed if we consider a single nanoparticle (heat point source) surrounded by infinite homogeneous medium, the temperature at the surface of the nanoparticle should be as $T = Q / (4\pi \cdot \kappa \cdot r) + T_{amb}$, with Q the heat power, κ the thermal conductivity of the surrounding medium and r the radius of the nanoparticle. I understand that the temperature, taking into account the collective heat effect of all nanoparticles, might be complex to estimate, but could authors comment at least on this effect? Also authors should speak about less heat energy instead of "lower temperature".

2 The method seems to be more efficient in term of energy used compared to the standard one with hot plate. What about the efficiency in term of chemical amount (iron, silver and silver)? Could authors estimate the nanoparticle synthesis yield for the full process of synthesis and compare with a standard approach?

3 Last sentence of the first paragraph of main part: "In our present work, we aimed to establish a colloidal synthesis of nanoparticles by utilizing AuBPs to generate the temperatures necessary." This sentence sound wrongly formulated, could authors correct it?

4 Second paragraph of main: "with elevated temperature are often needed", could authors give a range of temperature?

5 The article describes a method for IONPS, silver and palladium nanoparticle synthesis and should therefore mention some applications involving such nanoparticles in the introduction.

6 In Supplementary Tables 1-6: In the nanoparticle columns I don't understand when it is written "nanoparticle", should it be "monodisperse" ?

7 According to Supplementary Tables 1-6, the minimum temperature to achieve IONPs synthesis is 170°C instead of the 180°C mentioned in the first paragraph of Light-induced synthesis of IONPs part.

8. In the second paragraph of Light-induced synthesis of IONPs part, authors say that when increasing the temperature, the tips of AuBP "prone to degradation", they seem to degrade even at 160°C. This could give a clue to the author about the temperature at the vicinity of the NP (Gold nanoparticles start to melt around 200°C). As the method part mentions that the gold nanoparticles can be used several time, could also authors comment on consequence of tips deterioration with the absorbance spectrum?

9. The method for nanoparticle synthesise says that the solution should be heated at "inert condition", Why? Is it to avoid aggregation? Would it be different if we stir the solution?

10. Still in the method, the protocol for the IONP synthesis requires a bar magnet to separate the AUBP from the IONP but not for the silver. Is it because the silver is not magnetic? Is the separation less efficient without the magnet step? For the palladium it is written that a similar technique has been used, but it is not mentioned if the magnet is necessary.

11. There is no scale bar in the video.

We would like to sincerely thank all reviewers for investing the time and effort in reviewing this manuscript. We believe that the comments made by all five reviewers have given us fundamental insight and significantly helped to improve our work. Hopefully, in our present reply and revised manuscript, we manage to clarify misunderstandings and reinforce some of our hypotheses. Please find below a thorough point-by-point response to all the reviewers' comments and suggestions.

Point-by-point responses of reviewers' comments:

Reviewer #1 (Remarks to the Author):

The work "Unlocking the power of plasmonic photothermal activation to custom colloidal synthesis" by Biswas et al. shows an interesting wet chemistry synthesis protocol to obtain nanoparticles that typically are obtained by heating up the reaction solution, but instead using the photothermal properties of gold nanoparticles, in this case using silica-coated nano-bipyramids. With this protocol, they synthesized iron oxide, silver, and palladium nanoparticles. This approach is not new, as plasmonic nanoparticles have already been previously used to synthesize other nanoparticles or to grow them with initial seeds with the same or other materials (see for instance, ACS Nano 2018, 12, 8447–8455, J. Am. Chem. Soc. 2017, 139, 19, 6771–6776, J. Phys. Chem. C 2018, 122, 50, 28901–28909, Nature Materials (2016) 15, 889–895). In this work, the authors synthesize different types of nanoparticles and claim that with this approach they can decrease the temperature of the reaction with respect to the standard synthesis. The system is not very practical, as it requires the previous synthesis of the complex nanoheaters (which require several steps and some experience in the synthesis of nanoparticles), and lacks the simplicity of just using a hotplate. I am saying this because, they are using only the photothermal effect, but not other interesting properties of the plasmonic-driven reactions like the generation of hot electrons, or electromagnetic hotspots (the silica surrounding the gold core, most probably prevents it). Despite this, the approach used here could still be interesting as they obtained other types of nanoparticles, such as the core-satellite structures (Au@SiO₂-FeOx) that could be of interest for multifunctionality in different applications.

Regarding the manuscript I have not found any clear flaw in it, but I have a few questions/comments.

Response 1-general summary: We thank reviewer #1 for finding our work interesting and taking the time to give insightful comments and suggestions that have helped our work become more complete. Taking into consideration the mentioned references and other studies in the field, we hold a differing view from the reviewer concerning the novelty of our work. Previous studies, including the cited literature, do not deliberately use the plasmonic photothermal effect, but instead treat it as an unwanted side effect. They primarily rely on hot charge carriers generated by the particles to advance nanoparticle formation, thus enabling the synthesis of metallic nanostructures that are otherwise prepared under ambient conditions with a reducing agent. In contrast, our study demonstrates the application of the photothermal effect to successfully derive colloidal nanostructures as an alternative heating method, offering several advantages discussed in the manuscript. Moreover, while previous reports utilizing hot charge carriers show only metal particle formation, our work demonstrates metal oxide formation which is afforded through a distinct mechanism and elevated temperatures of 250–300 °C. In addition, the photothermal approach provides access to intricate assemblies and hybrids unattainable by conventional methods or hot charge carrier chemistry. We definitely agree with the reviewer's suggestion to incorporate other aspects of plasmonic materials, but unfortunately this is at

the moment beyond the scope of work. Regrettably, our perspectives on the novelty of the photothermal approach do not align, however, we respect reviewer #1's view and hope our revisions may have clarified our opinion.

Regarding the synthesis of SiO₂@AuBPs, we have developed a protocol enabling us to achieve a high yield, ~500 mL 20-30 OD in a single batch of synthesis allowing for more than hundreds of photothermal reactions. Additionally, these nano heaters are remarkably stable for more than six months without significant changes in their LSPR (response fig. 1). Thus, the complication of initiating the reaction with SiO₂@AuBPs is minimized. As for the complexity of utilizing light, we use a simple LED without the addition of any lens, controlled by a custom, user-friendly LabVIEW program, which in our opinion is not much more complicated than using a hotplate. We have added this now to the supplementary information (supplementary fig. 1) for the benefit of our readers.

Response Fig. 1 | Transmission electron microscope (TEM) image, and UV-Vis absorption spectrum. TEM images of silica encapsulated AuBP₈₅₀, **a**, before, and **b**, after more than six months of storage under 4 °C in ethanol, exhibiting excellent colloidal stability as observed from the **c**, UV-Vis spectrum.

1. In the synthesis of the nanoparticles, they did a double growth of the silica shell, first as porous and then as a continuous shell. Why is that? What is the function of the mesoporous inner shell?

Response 1-1: The internal mesoporous silica coating helps maintain a homogenous dispersion in addition to improving heat dissipation into the surrounding medium, resulting in rapid heating of the solution. An additional external continuous shell offers further chemical and thermal stability in different solvents, preventing the nanoparticles from reshaping via oxidative etching (response fig. 1). To highlight the importance of silica encapsulation we photothermally heated a DMF solution with non-encapsulated AuBPs. TEM images prior and subsequent to heating showed a dramatic influence on nanoparticle structure, that was manifested in the temperature profile with a sharp decrease in temperature after 5 minutes of photothermal heating (response Fig. 2). We thank reviewer #1 for pointing out this was not clearly stated in the original manuscript, we have now revised our manuscript accordingly (page no. 3, line no. 114-117, references 35-37), and supplementary information (page no. 6, supplementary fig. 10).

Response Fig. 2 | **a**, TEM image of non-encapsulated AuBP₈₅₀ prior to photothermal activation. **b**, Images taken of the AuBPs after conducting the photothermal heating 4 OD in DMF at 850 nm, which failed to reach or maintain the set photothermal temperature of 140 °C under conditions similar to response Fig. 8c.

2. The quantities of nanoparticles used for the synthesis are given in Optical Densities. What is the wavelength used to calculate this OD? The OD, could be useful to measure the photothermal capacity of the reaction solution, however, it would be useful (from the reproducibility point of view) to know the concentration in terms of Au moles and in terms of moles of nanoparticles (at least an estimation).

Response 1-2: The OD was calculated based on the maximum absorbance of the nanoparticles, for the present work an LSPR of the AuBPs at ~850 nm. Following reviewer #1's suggestion, a correlation between OD and the concentration of the AuBPs is now provided in supplementary fig. 49, which corresponds to ~19 ppm of Au for a 1 OD solution i.e., ~0.1 mM Au. In addition, the concentration of AuBPs in the same solution was calculated based on these results and was found to be ~10 pM (methods, page no. 8).

Response Fig. 3 | ICP-OES based correlation between OD and the concentration of gold, for a 1 mL, 1 OD solution of AuBP₈₅₀, the concentration of Au is ~19.2±0.1 ppm.

3. In line with the previous comment. The authors say on page 3, that the temperature of the reaction could be changed by changing the power or the concentration of nanoparticles. I guess that as the heat is higher at the surface of the nanoparticles, the more nanoparticles, the lower their temperature should be (as there is more surface available), so the results will depend on the nanoparticle's concentration. Have the authors tested any of the syntheses varying the nanoparticle concentration?

Response 1-3: We thank reviewer #1 for the thoughtful comment, varying AuBP concentration is indeed an interesting experiment absent from our original manuscript. To address this we used 15-25

OD of the AuBPs for the photothermal synthesis of iron oxide nanoparticles (~10 nm) at 200 °C (response fig. 4, and supplementary fig. 31), less than 15 OD has inconsistency in reaching such a high temperature. An increase in yield could be observed at higher OD, possibly the overall increase in silica surface area leads to more nucleation sites enabling higher yields. We did not observe an influence of the effect reviewer #1 mentioned concerning higher local temperatures for lower AuBP concentrations. This may be due to local temperatures being well over the necessary nucleation temperature regardless of the concentrations, thus an increase in the surface temperature would have little to no effect. Another possibility may be that the effect does have an influence, but the increase in surface area has a more dominant role in the formation of nanoparticles, overshadowing the difference in surface temperatures. An interesting follow up project could be to try and directly observe whether the concentration of plasmonic nanoparticles has an effect on the temperature in their immediate surrounding.

Response Fig. 4 | Iron oxide nanoparticle synthesis at a photothermal temperature of 200 °C with different concentrations (15-25 OD) of AuBPs with a slightly greater yield at higher AuBP concentrations.

4. As the photothermal effect at the nanoparticle surface seems to be the key for the synthesis at lower temperatures, could the authors estimate what temperature that would be? (I mean in comparison with solution temperature).

Response 1-4: Although this is a very interesting aspect of plasmonic photothermal heating a direct measurement of the nanoscale temperature is beyond the scope of the present study. There has been an effort to develop different strategies of nanothermometry and these studies demonstrate the temperature near the surface can reach as high as 200-300 °C in solution (response references 1-3). With that said, the nucleation temperature of different nanoparticles in the present work can provide an indication of the nanoscale temperature in comparison to the overall temperature of the solution. For example, please see response reference 4. We thank reviewer #1 for the idea.

5. The synthesis of iron oxide at the silica surface and then detachment seems quite interesting, however, I have difficulties imaging this happening for the Ag nanowires, how does this happen? From what is appreciated in the TEM images it looks like both nanowires and nanospheres are obtained in the reaction, and then the nanospheres end up sticking to the silica.

Response 1-5: Anisotropic growth of the Ag nanowires happens via seed-mediated growth through efficient polyol reduction in the presence of PVP. The nucleation step is primarily affected by the photothermal heating, similarly to our suggestion for the formation of other nanostructures, we

hypothesize that nucleation occurs on the surface, subsequently seeds are detached, and anisotropic growth is possible. During the growth of the nanowires they are most likely detached from the AuBPs possibly due to steric complexity as reviewer #1 rightly points out. To assist in clarification please see response fig. 5 showing an unpurified intermediate of the mentioned reaction. The seeds that are not detached from the surface do not undergo the growth process and can be seen attached to silica surface. We have mentioned this discussion in the revised manuscript, please see page 6, line no. 257-258.

Response Fig. 5 | TEM image of an unpurified reaction mixture during AgNW synthesis.

Reviewers #2 and #3, who co-reviewed the work (Remarks to the Author):

The authors describe a methodology to synthesize different colloidal nanoparticles, such as iron oxide, silver, and palladium, using gold bipyramids as plasmonic heaters capable of inducing nanoparticle formation via a photothermal process. One of the key aspects is the stabilization of gold bipyramids with a silica shell to stabilize them in different organic solvents where the nanoparticle growth occurs. Notably, the authors demonstrate that photothermal temperatures between 50 and 100 °C lower than those using estándar heating approaches are required for the synthesis processes. Furthermore, a mechanism based on the nucleation of nanoparticles on the silica surface and detachment during the late stages of the synthesis is proposed in the case of the iron oxide system. Although the concept of plasmonically mediated heating for the synthesis of colloidal nanomaterials is interesting, I think the quality of the obtained nanoparticles (iron oxide, silver, and palladium) is not sufficiently high compared with many current protocols found in the literature.

Response 2 & 3-general summary: We thank reviewers #2 & #3 for taking interest in our work, their insightful comments and insistence on important control experiments have strengthened our study. We concur with reviewers #2 & #3 that protocols with superior quality exist in current literature, however these procedures usually include complex reaction conditions and exotic reagents that could interfere with finding the effect of the photothermal method. In our work, we specifically chose and designed synthesis protocols that are both straightforward and free from toxic reagents, so that we could isolate the effect of the heating method on the reaction. We perceive elevating the level of complexity and thus of nanoparticle quality as a future task where we could build on the firm foundations that we have developed with our current research. We believe the methodology we

present can serve as a stepping stone for the research community to develop a library of nanoparticles with meticulous precision by utilizing light, discovering a wide range of intricate hybrid assemblies.

In addition, important control experiments are missing, and some questions remain open. For instance, what would happen if the synthesis of nanoparticles is attempted in the absence of gold bipyramids but in the presence of silica nanoparticles of similar dimensions?

Response 2 & 3-first comment: We thank reviewers #2 and #3 for bringing to our attention this important control experiment missing from our original work. Following their suggestion we have synthesized silica nanoparticles of comparable dimensions and tested their photothermal heating ability in the different solvents used throughout this work (response fig. 6). The results indicate the silica nanoparticles alone have no photothermal activity reaching temperatures well below those needed to initiate nucleation of the nanoparticles, clearly ruling out the possibility of nanoparticle formation in the absence of AuBPs. We have now added this to the supplementary information (supplementary fig. 2) for the benefit of our readers.

Response Fig. 6 | a, TEM image of silica nanoparticles with size ~150-200 nm and **b,** temperature profile in presence of silica nanoparticles (1 mg/mL) in different solvents demonstrating insignificant contribution to the photothermal temperature under 850 nm light irradiation.

The plasmon band of gold bipyramids is located at ca. 850 nm before irradiation experiments, resonating with the LED light source. However, after 45 min, the bipyramid reshaping shifts the band to 650 nm. Since the synthesis of iron oxide nanoparticles requires at least 4 h, how is the heating of the solution maintained when the plasmonic band is, after 45 min, off-resonance with the light source?

Response 2 & 3-second comment: We Thank reviewers #2 and #3 for their thoughtful comment, an explanation for this is indeed missing from our original work. As stated, initially the absorption of AuBPs starts at 850 nm and during the reaction slowly blue shifts to around 650 nm after 45 minutes. This shift is produced by the blunting of the bipyramidal tips, which occurs at high temperatures (response fig. 7a-b). The temperature can be maintained high despite the AuBPs being off-resonance due to weak absorption around 850 nm (response fig. 7a,c), without contribution to heat from any other sources. This is likely because, once the set temperature programmed by LabVIEW is reached, despite the fact that the bipyramids undergo a gradual reshaping process, they still need to adjust to a slight temperature change (less than 2 °C) to maintain stability. In order to further validate this observation, we repeated the experiments at 200 °C with SiO₂@AuBP₈₅₀ and nanoparticle precursor for 4 h, subsequently, the mixture was allowed to cool down to room temperature, followed by exposure to light irradiation under the same conditions (response fig. 7d). To our expectations, the

AuBPs were no longer able to attain the high temperature of 200 °C. However, they were able to reach a maximum temperature of ~160-165 °C, albeit at a much slower rate, and sustain this temperature for an additional 4 hours. To further reinforce the observations, we synthesized AuBP₆₆₀ and compared photothermal heating performance to that of AuBP₈₅₀ under 850 nm light irradiation (response fig. 8). We observed that AuBP₆₆₀ managed to reach a temperature closer to the targetted temperature under the same conditions, with a much slower rate compared to AuBP₈₅₀. On the other hand, AuBP₈₅₀ nanoparticles without a silica shell were found to be completely degraded under the same photothermal heating conditions, failing to attain or maintain the desired temperature (response fig. 2). We have added these now to the supplementary information (supplementary fig. 8-10) for the benefit of our readers.

Response Fig. 7 | **a**, Temperature profile of SiO₂@AuBP₈₅₀ in octadecene-oleic acid-oleylamine mixture without iron oleate under 850 nm light irradiation. **b**, TEM image after 60 min of reaction showing degradation of the sharp tips of AuBPs keeping overall structural stability intact. **c**, UV-Vis absorbance spectrum of the corresponding nanoparticles showing shifting off of the LSPR to a lower wavelength. **d**, Temperature profile of SiO₂@AuBP₈₅₀ with iron oleate in octadecene-oleic acid-oleylamine mixture followed at 200 °C for 4 h. Subsequently the LED was turned off, and the reaction mixture was cooled at room temperature. The LED under the same condition was turned on afterward, exhibiting the AuBP nanoparticles' heating ability slowed down and could reach only ~160-165 °C. The temperature however remained consistently maintained high for 2-4 h throughout for both the reactions (**a**, **d**) due to weak absorption of AuBPs around 850 nm, and likely because once the set temperature programmed by LabVIEW is reached, despite the fact that the bipyramids undergo a gradual reshaping process, it's easier to maintain stability.

Response Fig. 8 | **a**, TEM image of SiO₂@AuBP₆₆₀ with corresponding **b**, UV-Vis absorption spectrum. **c**, Temperature profile of 4 OD silica-coated AuBPs in DMF under 850 nm light irradiation for 30 min demonstrating slow heating profile of AuBP₆₆₀ compared to AuBP₈₅₀ which failed to reach a set temperature of 140 °C.

A 100 W light source focused on a small vial, such as those used for the synthesis, implies a high amount of energy on a small volume. Is it possible that the solvent or reactant interaction with such a light source promotes the nucleation and growth of iron oxide (silver and palladium also) nanoparticles?

Response 2 & 3-third comment: We greatly appreciate reviewers #2 & #3 for pushing us to carry out important control experiments, reinforcing our claims throughout the manuscript. Firstly, we assessed the actual power output of our so called 100 W LEDs, and found them to provide an actual power of 6-7 W/cm² (supplementary fig. 3), which is relatively high but as we later demonstrate not enough to produce considerable heat without a photothermal agent. Next, we ran controls for each nanoparticle formation reaction without AuBPs under identical conditions. In the case of IONPs and AgNPs, the reaction mixture could not reach the nucleation temperature and certainly no nanoparticle formation was observed (supplementary fig. 2, 7, and 39).

In the case of palladium, when toluene was used we obtained similar results to IONPs and AgNPs, where the reaction mixture excluding AuBPs did not show an increase in temperature and no nanoparticles formed. For the nanoparticle synthesis in ethanol, we found that the control reaction without AuBPs can produce aggregated nanoparticles under light irradiation, having some resemblance to the self-assemblies afforded from the reaction with AuBPs. When the reaction

solution was irradiated, a rise in the temperature after 10 minutes of illumination was noticed, signifying the emergence of Pd nanoparticle seeds. We hypothesize that these seeds, now serving as photothermal agents, play a crucial role in initiating nucleation as described in the manuscript for the AuBP initiated reaction and producing aggregation of nanoparticles (response Fig. 9). Importantly, we also found that in an ethanolic medium, even at room temperature nanoparticles formed over time. In this context conducting control experiments without AuBPs under NIR irradiation provided us with an additional understanding of the self-assembly process.

Again, we thank reviewers #2 & #3 for this comment as it led us to better understand the formation of well-defined Pd assemblies with photothermal methodology. We have now added a discussion to the revised manuscript regarding our new findings (page no. 7, line no. 292-302).

Response Fig. 9 | a-c, Temperature profile **a**, and UV-Vis absorption spectra **b-c**, (in toluene **b**, in ethanol **c**), for control reactions only without gold bipyramids in different solvents for photothermal synthesis of palladium nanoparticle under 850 nm light irradiation. In toluene, insignificant contribution of the reaction mixture/solvents towards photothermal synthesis of palladium nanoparticles, and notably, no nanoparticle was isolated without AuBPs. **d**, Digital images showing stability of the palladium precursor in different solvents for different time intervals. **e-f**, TEM images of palladium nanoparticle isolated from the reaction in ethanol which was kept at room temperature for a week **e**, (d-vi) and synthesized under photothermal condition without AuBPs under light **f**.

Could it be that free radicals are generated during the irradiation process (in the presence and absence of gold bipyramids) and participate in nanoparticle growth?

Response 2 & 3-fourth comment: Based on the literature reports on the selected conventional reactions and the control reactions conducted under photothermal conditions (where no nanoparticles were formed in the absence of AuBPs and with AuBPs below 160 °C), we can confidently rule out the generation of free radicals possibly involved in the reaction. Furthermore, we employed a robust silica encapsulation around the AuBPs with inert reaction conditions maintained throughout the experiments which further rule out any major contribution of free radicals.

In my view, the scalability of the proposed synthesis methods is limited by the need to concentrate a powerful light source on a small volume of growth solution. Moreover, the amount of gold by pyramids is quite high. An OD of 15-20 implies a concentration of Au between 2.5 and 3.5 mM, which is quite demanding. Do the authors have a potential solution to these issues?

Response 2 & 3-fifth comment: We understand the concern of reviewers #2 & #3. In line with the previous response (response 2 & 3 - third comment), we have now measured the actual power output given from our LED to be 6-7 W/cm². Additionally, we employed a low-cost LED source readily available from online retailers for just a few tens of dollars and we do not use any concentrating lens whatsoever (supplementary fig. 3). Unfortunately, we do not have at our disposal LED chips with a larger surface area, but we are confident that scaling up our current synthesis models is a matter of suitable equipment as is true for conventional heating methods.

Following the reviewers' suggestions, we have established a correlation between the optical density (OD) and the concentration of the AuBPs (supplementary fig. 49, and response fig. 3). Our results show that a concentration of 1 OD corresponds to only ~19 ppm of Au, thus if we consider a reaction carried out with 20 OD AuBP, the concentration of Au is of ~380 ppm. This concentration of gold is comparable with catalyst concentrations in highly efficient industrial processes (response reference 5). Moreover, the synthesis of AuBPs can be done at large scale as pointed out previously (response 1-general summary), and the nanoparticles can be easily stored for months if not years (supplementary fig. 1).

Finally, the low temperatures and milder reaction conditions of the photothermal approach offer a technical advantage that together with the described reasons lead us to believe that our overall approach can be made cost-effective for future scale-up endeavors.

More importantly, the discussion includes the concept of "catalysis". However, it should be clarified why the described photothermal synthesis can be considered as a catalytic process and not simply as a heterogeneous nucleation process, for instance there is not deep explanation about the reactivity mechanisms involved in the process.

Response 2 & 3-sixth comment: We thank the reviewers for bringing to our attention that we were not clear on this issue. We referred to the AuBPs as catalysts, because when carrying out the reaction in their presence we observed lower nucleation temperatures while achieving higher yields (response fig. 13). We thought of this as lowering the activation energy of the reaction without altering the reactants or product i.e., without changing the Gibbs free energy. Additionally, we noticed a slight improvement in the overall yield as the concentration of AuBPs increased (response fig. 4). Taking these observations into account, we believe that the term "catalytic behavior" aligns well with our discussion. This is not to say that the term heterogeneous nucleation process does not also fit in this case. Thus, to avoid any confusion, based on the reviewers' suggestion, we have modified the manuscript excluding the term catalysis throughout.

Reviewer #4 (Remarks to the Author):

The paper report the use of plasmonic Au nanopyrimids coated with SiO₂ as a heater to heat a reaction solution of nanoparticle precursors to form the desired nanoparticles. Examples on light-induced synthesis of iron oxide, silver, and palladium nanoparticles were given. The photothermal method was also extended to synthesize Ag nanowires, suggesting its potential to achieve anisotropic growth of

nanostructures. The reported photon driven nanoparticle synthesis is of some interest, but the synthesis outcome is not very competitive to the well-established thermal synthesis method. The work does not show high enough novelty or importance for this journal.

Response 4-general summary: We thank reviewer #4 for constructive comments and suggestions, motivating us to strengthen our claims and improve this study. To avoid being repetitive we direct the reviewer to response 2 & 3-general summary where we elaborate on our opinion as to the competitiveness of the nanoparticles afforded by the photothermal approach. We have now thoroughly revised our manuscript taking into account all reviewer comments and suggestions, we believe this version is more comprehensive and complete and hope it now meets this journal's novelty and impact standards.

It is not clear why a special feature of nanostructured Au, Au bipyramids (AuBPs), was selected as a plasmonic heater. What is unique part of these bipyramids over Au nanorods? The justification given by the authors "as they exhibit tunable narrow localized surface plasmon resonance (LSPR) bands and can be synthesized with high yields.10-12" does not seem to justify the uniqueness and importance of this selection as Au nanorods with similar optical properties have been prepared in large scale.

Response 4-first comment: We agree with the reviewer's viewpoint. However, we still believe as stated that the LSPR wavelength of AuBPs can be easily tuned by adjusting the ratio of gold chloride to seed nanoparticles. Additionally, the synthesis affords highly monodisperse structures in high yield that together with the intrinsic properties of the bipyramidal geometry result in narrow absorption peaks. The combination of tuneable and narrow LSPR peaks enables limiting the aggressive photothermal response to a minimal range of light frequencies. We considered this to be a unique advantage. This discussion is added in the revised manuscript Introduction page no. 2, line no. 56-61.

As pointed out by the reviewer, we agree that the photothermal methodology can undoubtedly be carried out using AuNRs, clarification of this is presented in response fig. 10-11 by utilizing SiO₂@AuNR as photothermal agents in the light induced synthesis, affording similar results to the ones obtained with AuBPs. Additionally, we synthesized SiO₂@AuBP₆₆₀ which can be used at a visible wavelength of 660 nm yielding similar results (please see response fig. 12) to highlight the described wavelength tunability. These experiments demonstrate the versatility of our current approach, showcasing its applicability across various plasmonic nanostructures and light frequencies. We have now added this to the revised manuscript page no. 4, line no. 167-171 for the benefit of our readers.

Response Fig. 10 | **a-b**, TEM images of silica-coated gold nanorods ($\text{SiO}_2\text{@AuNR}$) at **a**, low, and **b**, high magnifications. **c**, Normalized UV-Vis absorption spectra of silica-coated AuBP and AuNR, and **d**, corresponding temperature profile under 850 nm light irradiation.

Response Fig. 11 | **a-b**, TEM images of **upper panel**, iron oxide nanoparticles (IONPs, $\sim 4\text{-}6$ nm), and **lower panel**, corresponding IONP@AuNR_{850} (IONPs, $\sim 5\text{-}7$ nm) at different magnifications, synthesized with photothermal heating at 180°C for 2 h under 850 nm light irradiation. **c**, **upper panel**, Temperature profile of the reaction, and **lower panel**, UV-Vis absorption spectrum of the IONP@AuNR with magnetic response (inset).

Response Fig. 12 | **a-b**, TEM images of **a**, iron oxide nanoparticles (IONPs, ~4-5 nm) and **b**, corresponding IONP@AuBP₆₆₀ (IONPs, ~5-6 nm), synthesized with photothermal heating at 180 °C for 2 h under 660 nm light irradiation. **c**, Temperature profile for the photothermal synthesis. **d**, UV-Vis absorption spectrum of the IONP@AuBP₆₆₀ with magnetic response (inset).

The main novelty claimed by the authors is that using the plasmonic heating can lower the reaction temperature for nanoparticle synthesis. This is certainly interesting, but the degree of the temperature drop demonstrated in this paper is not that significant. This, plus the complexity introduced by Au nanopyrimids synthesis, SiO₂ coating, and light irradiation, does not provide much advantage over the conventional thermal heating approach to nanoparticle synthesis.

Response 4-second comment: We thank reviewer #4 for taking interest in our work. We respect the reviewer's point of view on the degree of temperature difference between the photothermal method and conventional methods. However, in our opinion decreasing the temperature by up to 100 °C is quite significant in terms energy efficiency and facilitates technical aspects of the synthesis. For the photothermal synthesis of IONPs the temperature used is well below the boiling point of the reaction mixture avoiding evaporation followed by condensation and splashing, a common problem in high-temperature colloidal synthesis. Furthermore, the absence of an external heat source allows for greater control over the reaction temperature, enabling a deviation of less than 2 °C for temperatures exceeding 150 °C. In our experience, lowering the reaction temperature and gaining better control over it greatly increases its reproducibility. In addition, as reviewer #4 suggested, we have now shown a clear advantage in reaction yields for the photothermal approach (response fig. 13), further improving the efficiency of the photothermal process.

Regarding increasing the complexity of the reaction when using the photothermal method, we direct the reader to response 1-general summary where we address this issue.

In the synthetic section, there is no report on synthetic yields on nanostructures made by their light-induced synthesis. In the section “Catalytic behavior, and scope of the photothermal approach”, what catalytic behavior the authors are referring to?

Response 4-third comment: We agree with reviewer #4 that an analysis of product yields is missing from our original manuscript and thank the reviewer for pointing this out. We have now added a thorough analysis of reaction yields for all three types of nanoparticles (IO, Ag, and Pd), comparing photothermal reactions to identical conventional reactions. Our results show a clear advantage for the photothermal method in terms of product yield in all tested cases (response fig. 13). We have added these now to the supplementary information (supplementary fig. 31, 39, and 48) for the benefit of our readers. Regarding the use of the term catalysis to describe the behavior of the AuBPs, we kindly direct the reader to response 2 & 3-sixth comment, where we address this issue.

Response Fig. 13 | Synthetic yield of nanoparticles under photothermal and conventional reactions. **a**, Iron oxide nanoparticle synthesis for **i**, ~4-6 nm particles, **ii**, ~10 nm particles, **iii**, ~6-8 nm triangular particles between 180-250 °C. **b**, Iron oxide nanoparticle synthesis corresponding to **i**, ~2-3 nm particles, and **ii**, ~15 nm nanoplates at 300 °C. **c**, Iron oxide nanoparticle synthesis at a photothermal temperature of 200 °C with different concentrations (15-25 OD) of AuBPs with a slightly greater yield at higher AuBP concentrations. **d**, Silver nanoparticle, **e**, silver nanowire, and **f**, palladium nanoparticle synthesis in different reaction conditions exhibiting consistent and greater nanoparticle yield under photothermal conditions at lower temperatures similar to the iron oxide nanoparticles.

In the synthesis of Ag nanoparticles, why does the 250 μ L of FeCl₃ solution need to mix with 180 mg of AgNO₃ in order to prepare Au nanoparticles? What is the role of FeCl₃? Is there any Fe involvement in the final Ag nanostructures?

Response 4-fourth comment: We thank reviewer #4 for this interesting question. A minute amount of FeCl₃ solution added in the beginning aids in facilitating slow nucleation and growth of the nanowire possibly through AgCl formation by utilization of its Cl⁻ ions, the amount of free Ag⁺ ions in solution is decreased, and the kinetics of the reduction process of AgNO₃ are slowed down (response reference

6). We could not find any Fe involvement in the final Ag nanostructure as observed from the EDS spectra (response fig. 14).

Response Fig. 14 | EDS spectrum showing silver as the major element in the nanowire.

Reviewer #5 (Remarks to the Author):

Authors present an original method for monodisperse nanoparticles synthesis. Using a photothermal approach involving plasmonic effect of silica coated gold bipyramidal nanoparticles (AuBP), they successfully managed to synthesise iron oxide, silver and palladium NPs with different sizes and shapes. Besides, they studied the effect of different parameters of the protocol (temperature, exposure time, concentration of reagents) and showed that their approach requires less energy than standard approaches using a hot plate.

Response 5-general summary: We thank reviewer #5 for appreciating our work and describing it as an “original method” in the field. We have thoroughly considered the following comments which led to the improvement of this study.

Although I am supportive, I do have some critics:

1. Authors claim that they could synthesise NPs at lower temperature than standard approach. As an IR thermometer has been used, the temperature measured is rather an average temperature depending on resolution of the camera (which should be few μm). That is the reason why, I doubt that the temperature at the vicinity of the AuBP NPs is such lower. Indeed if we consider a single nanoparticle (heat point source) surrounded by infinite homogeneous medium, the temperature at the surface of the nanoparticle should be as $T=Q/(4\pi\cdot\kappa\cdot r)+T_{\text{amb}}$, with Q the heat power, kappa the thermal conductivity of the surrounding medium and r the radius of the nanoparticle. I understand that the temperature, taking into account the collective heat effect of all nanoparticles, might be complex to estimate, but could authors comment at least on this effect? Also authors should speak about less heat energy instead of “lower temperature”.

Response 5-1: We thank reviewer #5 for bringing to our attention that our work was not sufficiently clear. Our claim of lower temperatures refers to the bulk temperature of the solution, as this is what

we measure as the reviewer rightly points out, and not to the local heat energy surrounding the near vicinity of the particle. We regret that we are unable to measure the precise temperature near the surface of the nanoparticles in our case. However, there are reports highlighting that there exists a steep temperature gradient from the particle into the solution, where the particle acts as hotspot (response references 1-4). We agree with the reviewer's comment on the use of heat energy when considering the nano-environment around the particle, but as stated in this case we refer to the temperature of the entire solution, thus we believe using "lower temperature" is accurate.

2. The method seems to be more efficient in term of energy used compared to the standard one with hotplate. What about the efficiency in term of chemical amount (iron, silver and silver)? Could authors estimate the nanoparticle synthesis yield for the full process of synthesis and compare with a standard approach?

Response 5-2: We greatly appreciate reviewer #5's suggestion which has undoubtedly helped to improve our work. Please find response 4-third comment where we address this issue in full detail.

3. Last sentence of the first paragraph of main part: "In our present work, we aimed to establish a colloidal synthesis of nanoparticles by utilizing AuBPs to generate the temperatures necessary." This sentence sound wrongly formulated, could authors correct it?

Response 5-3: We thank reviewer #5 for the careful correction. The sentence is modified in the revised manuscript for better understanding as "In our present work, we aimed to generate the temperatures necessary for colloidal synthesis of nanoparticles utilizing photothermal activation of AuBPs."

4. Second paragraph of main: "with elevated temperature are often needed", could authors give a range of temperature?

Response 5-4: We have now mentioned the temperatures (300-380 °C) in the revised manuscript. We thank reviewer #5 for pointing this out.

5. The article describes a method for IONPS, silver and palladium nanoparticle synthesis and should therefore mention some applications involving such nanoparticles in the introduction.

Response 5-5: We thank reviewer #5 for the suggestion. We have now mentioned it in the introduction and added reference 29 to the revised reference list of the manuscript.

6. In Supplementary Tables 1-6: In the nanoparticle columns I don't understand when it is written "nanoparticle", should it be "monodisperse" ?

Response 5-6: Sorry for the confusion. We have corrected it as "monodisperse".

7. According to Supplementary Tables 1-6, the minimum temperature to achieve IONPs synthesis is 170°C instead of the 180°C mentioned in the first paragraph of Light-induced synthesis of IONPs part.

Response 5-7: This is indeed the case we thank reviewer #5 for correcting us.

8. In the second paragraph of Light-induced synthesis of IONPs part, authors say that when increasing the temperature, the tips of AuBP "prone to degradation", they seem to degrade even at 160°C. This could give a clue to the author about the temperature at the vicinity of the NP (Gold nanoparticles start to melt around 200°C). As the method part mentions that the gold nanoparticles can be used

several time, could also authors comment on consequence of tips deterioration with the absorbance spectrum?

Response 5-8: The sentence is modified by correcting the tips of AuBP “prone to degradation” to “In addition, under the synthesis condition, the tips of AuBPs were found to be degraded”. The SiO₂@AuBPs were also found to be degraded under conventional heating under the same reaction condition (supplementary fig. 22-23), and this observation is also noticed for other type of nanostructures such as SiO₂@AuNR₈₅₀ or SiO₂@AuBP₆₆₀ which indicates the chemical environment in the synthesis was primarily responsible for such degradation at high temperature (supplementary fig. 24-27).

As noted by the reviewer the SiO₂@AuBPs undergo an LSPR blue shift after the reaction, thus, reuse can be done at the shifted wavelength with a different LED source. Importantly, degradation is not unique to the bipyramidal structure but was also observed in rod shaped nanoparticles (please see response fig. 10-11).

9. The method for nanoparticle synthesise says that the solution should be heated at “inert condition”, Why? Is it to avoid aggregation? Would it be different if we stir the solution?

Response 5-9: Argon flow is, in general, a common practice in colloidal synthesis involving metal-carboxylate decomposition because the reaction temperatures may be above the flammable point of the organic compounds. Moreover, the inert condition has benefits over unwanted/uncontrolled oxidation leading to impurity/aggregation at high temperatures. Additionally, it rules out any oxygen-related free radical involvement in the reaction, and also the possibility of decomposition of solvents. Stirring was maintained for better mixing of the solution, although our system is near homogenous and to reduce superheating.

10. Still in the method, the protocol for the IONP synthesis requires a bar magnet to separate the AUBP from the IONP but not for the silver. Is it because the silver is not magnetic? Is the separation less efficient without the magnet step? For the palladium it is written that a similar technique has been used, but it is not mentioned if the magnet is necessary.

Response 5-10: The purification steps can be used efficiently as described without using an external bar magnet for all of the nanoparticles. However, with iron oxide particles the use of the external bar magnet for purification is an added advantage due to its magnetic property. Palladium and Silver nanoparticles are not magnetic and therefore their purification was done without a magnet. We hope we have managed to clarify this issue and thank reviewer #5 for taking the time to make sure our work is understood correctly.

11. There is no scale bar in the video.

Response 5-11: Scale bar is added.

Response references

1. Baffou, G., Polleux, J., Rigneault, H. & Monneret, S. Super-heating and micro-bubble generation around plasmonic nanoparticles under cw illumination. *J. Phys. Chem. C* **118**, 4890–4898 (2014).

2. Baffou, G. & Rigneault, H. Femtosecond-pulsed optical heating of gold nanoparticles. *Phys. Rev. B* **84**, 1–13 (2011).
3. Baffou, G., Cichos, F. & Quidant, R. Applications and challenges of thermoplasmonics. *Nat. Mater.* **19**, 946–958 (2020).
4. Cui, X., Ruan, Q., Zhuo, X., Xia, X., Hu, J., Fu, R., Li, Y., Wang, J. & Xu, H. Photothermal nanomaterials: A powerful light-to-heat converter. *Chem. Rev.* **123**, 6891–6952 (2023).
5. Mitchell, S., Qin, R., Zheng, N. et al. Nanoscale engineering of catalytic materials for sustainable technologies. *Nat. Nanotechnol.* **16**, 129–139 (2021).
6. Coskun, S., Aksoy, B., & Unalan, H. E. Polyol synthesis of silver nanowires: an extensive parametric study. *Cryst. Growth Des.* **11**, 4963–4969 (2011).

Reviewer comments, second round -

Reviewer #1 (Remarks to the Author):

The authors have answered satisfactorily to my previous comments. I would like to add two new comments.

First, I think they should rephrase the sentences where they claim that the process in terms of energy is more efficient. Efficiency is complex to evaluate in this process but surely, there are big energy losses in the current-to-light transduction in the LED and the light-to-heat conversion in the solution. Instead, the term lower temperature synthesis seems more correct.

Second, in many reactions, like in the case of IO nanoparticles, the resulting nanoparticles (size and polydispersity) not only depend on the reaction temperature but also on the initial temperature ramp. This is due to the complex process of nucleation and the interconnection between nucleation and growth in seedless reactions. The authors should perform a control experiment, at least for one of the syntheses, using the same heating ramp and maintaining the rest of the conditions identical, to discard the ramp effect.

Reviewer #2 (Remarks to the Author):

The manuscript reports on the investigation of a photothermal-mediated synthesis of different metal and metal oxide nanoparticles as nanoplasmonic heaters. The authors have properly answered many of the questions provided by the referees, but, although the development of new methodologies of synthesis of plasmonic nanoparticles is always an interesting topic, the presented results are not novel themselves, and from my point of view and considering the reviews of the other referees, they do not justify the publication in a very competitive journal such as Nature Communications. Therefore, I do question whether Nature Communications is the suitable outlet of this work, and I feel that it will be of interest to a rather specialized subset of authors in other journal.

Reviewer #3 (Remarks to the Author):

I appreciate the efforts the authors have dedicated to addressing our concerns and modifying the manuscript. However, important issues remain, such as the need for the synthesis of complex nanoheaters in large amounts (in my opinion, 380 ppm of Au needed for heating is quite a high concentration if we consider it is a precious metal) and the low size and shape dispersity control over the products (especially Ag and Pd nanoparticles). Moreover, these are intrinsic problems of the described photothermal synthesis method, which are difficult (or impossible) to address. For these reasons, I believe the manuscript is unsuitable for publication in Nature Communication.

Reviewer #4 (Remarks to the Author):

The authors tried hard to answer reviewers' questions and revised the paper accordingly. However, I am not convinced that the revised version shows improved quality compared to the original version. Worse, some of the new additions reflect the fact that the synthetic method they presented is of little use as a general method for nanoparticle synthesis.

Judging from the synthetic perspective, I do not feel the reported synthesis is an advance in the field as the nanoparticles made from the synthesis have poor size and shape distributions (The authors claim they have monodispersed nanoparticles, but this is not supported by TEM images they presented).

In answering the synthetic yield question, the authors did not give the yield numbers, rather, they gave a method how the synthetic yield should be calculated, which is itself questionable. To use

ICP-AES to analyze metal concentration from the reaction mixture and then to calculate synthetic yield can be very misleading. The fact that the authors cannot provide the real yield numbers suggest that they could not even separate any measurable amount of product from their synthesis started with hundreds of mg of metal precursors. The synthesis with such a low product output has no much use for others in the field.

The claimed assembly of nanoparticles on SiO₂-coated Au during the synthesis is also not supported. One would not agree to call the random disposition of nanoparticles on the SiO₂ surface a controlled assembly. A much-controlled assembly on SiO₂ can be done easily with well-established silane surface chemistry. Therefore, the claimed assembly method has no advantage to what has been known in the field.

Overall, although the plasmonic heating for synthesis is an interesting idea, I am not convinced that the authors have demonstrated a meaningful synthesis from this heating idea, and their claims made in the paper are not supported. Therefore, this manuscript/work is not suitable for Nature Comm.

Reviewer #5 (Remarks to the Author):

The authors addressed my concerns adequately.

We would like to sincerely thank all reviewers for dedicating the time and effort in reviewing this manuscript. We believe that the comments made by all five reviewers have given us fundamental insight and helped substantially to improve our work further. In our present response and revised manuscript, we have made diligent efforts to address any misunderstandings and reinforce our hypotheses. Please find below a thorough point-by-point response to all the reviewers' comments and suggestions.

Point-by-point responses of reviewers' comments:

Reviewer #1 (Remarks to the Author):

The authors have answered satisfactorily to my previous comments. I would like to add two new comments.

Response 1-general summary: We would like to express our appreciation for reviewer #1's positive feedback and recognition of our revisions to previous comments. Your concerns are addressed below.

1. First, I think they should rephrase the sentences where they claim that the process in terms of energy is more efficient. Efficiency is complex to evaluate in this process but surely, there are big energy losses in the current-to-light transduction in the LED and the light-to-heat conversion in the solution. Instead, the term lower temperature synthesis seems more correct.

Response 1-1: We understand reviewer #1's concern. We have now calculated the energy output for a photothermal setup, and a conventional setup (using an IKA plate) to heat a solution under identical reaction conditions. Clearly, the photothermal approach demonstrated high energy efficiency, utilizing 28,000 Joules, while the conventional heating required 360,000 Joules for the same reaction at 200 °C examined for 10 min (supplementary fig. 3). In addition to the energy output, as indicated by reviewer #1, we want to mention as a reactivity (or temperature) unattainable by conventional heating can be achieved with the photothermal method, and when considering energy efficiency, the photothermal activation is favorable where the mixture itself contains the heat source, therefore only the solution is being heated in contrast with an oil bath or an oven where the heat source, oil or air in the oven and the solution are being heated. Additionally, the enhanced reactivity enables the use of lower reaction temperatures or shorter times, further cutting the energy cost of the process. These are significant advantages for these reasons we strongly believe that the energy-efficient synthesis correlates well with the presented concept.

2. Second, in many reactions, like in the case of IO nanoparticles, the resulting nanoparticles (size and polydispersity) not only depend on the reaction temperature but also on the initial temperature ramp. This is due to the complex process of nucleation and the interconnection between nucleation and growth in seedless reactions. The authors should perform a control experiment, at least for one of the syntheses, using the same heating ramp and maintaining the rest of the conditions identical, to discard the ramp effect.

Response 1-2: Wow! We would like to thank reviewer #1 for pointing out a crucial point that we initially overlooked. In response to your valuable suggestion, we conducted control reactions using a conventional heating method with heating ramps of approximately 36 °C/min at 210 °C and ~22 °C/min at 250 °C, which simulate the ramps observed under photothermal synthesis conditions. These

control reactions were then compared with regular heating without any programmed ramp effect (please refer to supplementary figure 29). As correctly pointed out by reviewer #1 heating ramps are crucial in many cases, however, we did not observe any significant ramp effect on the nanoparticles' morphology. This reinforces the idea that higher temperatures in the vicinity of the plasmonic heaters enable us to achieve an overall lower synthesis temperature. We extend our sincere gratitude to reviewer #1 for investing the effort in ensuring our manuscript is free of error.

Reviewers #2 & #3 (co-reviewed the previous version):

Reviewer #2 (Remarks to the Author):

The manuscript reports on the investigation of a photothermal-mediated synthesis of different metal and metal oxide nanoparticles as nanoplasmonic heaters. The authors have properly answer many of the questions provided by the referees, but, although the development of new methodologies of synthesis of plasmonic nanoparticles is always an interesting topic, the presented results are not novel themselves, and from my point of view and considering the reviews of the other referees, they do not justify the publication in a very competitive journal such as Nature Communications. Therefore, I do question whether Nature Communications is the suitable outlet of this work, and I feel that it will be of interest to a rather specialized subset of authors in other journal.

Response 2-general summary: We thank reviewer #2 for appreciating the fact that we were able to answer satisfactorily the concerns raised previously. Reviewer #2's positive view of our work in developing new methodologies using the plasmonic photothermal effect is valued.

We would like to emphasize some critical findings from our designed research. As known conventionally, the photothermal approach was thought about merely as a side effect of plasmonic processes, with limited efforts made by researchers to isolate and exploit this effect for diverse applications. Previous research mainly focused on biomedical and catalytic applications, where the full potential of the plasmonic photothermal effect in solution remained largely untapped, whereas lithography techniques were advanced later by slightly improving the capability albeit not truly in solution. These impediments were largely due to having no control over the heat generated by the plasmonic materials. Our study aims to address this gap and proposes a solution-based approach where we can precisely control the temperature via a customized labview program to unlock the untapped potential of the photothermal effect demonstrated through achieving high-temperature (250-300 °C) synthesis of colloidal nanoparticles. By advancing the photothermal effect in colloidal synthesis, we demonstrate that conventional heating methods are limited compared to the possibilities offered by our approach. Notably, our research showcases the energy efficiency and mild reaction conditions achievable through this process. Moreover, our work put on display the versatility of the approach, as we successfully employed various plasmonic nanoparticles, different wavelengths of light, and demonstrated the high-yield synthesis of a variety of nanoparticles. Additionally, our method allows for the controlled synthesis of nanohybrids (metal or metal oxide) or assemblies through simple light irradiation operando. We firmly believe that these results represent a novel and significant achievement in the research areas related to plasmonic photothermal processes. Our findings not only expand the understanding of the potential offered by the plasmonic photothermal effect, but also showcase results that hold great promise for various applications in diverse fields.

We are confident that the broader audience of *Nature Communications* will appreciate the importance of our work and will be benefitted from applying this concept in their own research, discovering reactions previously unattainable through conventional heating methods.

Reviewer #3 (Remarks to the Author):

I appreciate the efforts the authors have dedicated to addressing our concerns and modifying the manuscript. However, important issues remain, such as the need for the synthesis of complex nanoheaters in large amounts (in my opinion, 380 ppm of Au needed for heating is quite a high concentration if we consider it is a precious metal) and the low size and shape dispersity control over the products (especially Ag and Pd nanoparticles). Moreover, these are intrinsic problems of the described photothermal synthesis method, which are difficult (or impossible) to address. For these reasons, I believe the manuscript is unsuitable for publication in Nature Communication.

Response 3-general summary: We would like to express our appreciation for reviewer #3's positive feedback and recognition of our efforts to fully address previous comments. Without being repetitive about the synthetic scalability of the AuBPs, as we have already cleared our viewpoint previously, we want to highlight that when considering the presented photothermal synthesis of the hybrid materials (metal or metal oxide @AuBPs), surely the concerns raised on the amount of gold are irrelevant. Additionally, in the synthesis of Palladium nanoparticles, we show that a gold concentration of 40 to 80 ppm is sufficient to advance the reaction.

In response to reviewer #3's comment, we have now improved the morphology control of silver nanoparticles by slightly adjusting the synthesis condition (80 °C temperature difference compared to conventional heating) and employed palladium nanocube synthesis through photothermal reactions demonstrating less than 5% standard deviation in their morphology (response fig. 1). Our NPs are now comparable, if not better than high quality commercially available NPs from Sigma Aldrich, the obtained quality seems to be competitive. We have added this as supplementary fig. 49 for the benefit of our readers.

In his remarks, reviewer #3 states that lowering the amount of gold in the photothermal reaction and improving on shape and size monodispersity are intrinsic issues that might be impossible to resolve. In our opinion this is not the case. First, in just a few weeks since we started working on the current revision, we have already managed to improve the control over size and shape dispersity. Thus, it is reasonable to think that with enough effort one may produce NPs of any kind with very high monodispersity and that this is not an intrinsic unsolvable problem of the photothermal methodology. Second, as for the amount of gold, it is reasonable to think that a more specialized set up concentrating a greater number of photons on the solution may lead to obtaining high temperatures with lower gold concentrations. Additionally, we have shown that the method is not specific to a given structure or wavelength, thus one could hypothesize that other plasmonic NPs such as Cu or Ag that have strong photothermal response and are much more economical, may also lead to similar results. The presence of a plasmonic agent is an intrinsic property of our system, but whether its concentration is too high is in itself debatable and lowering it is definitely not an impossible task.

These findings significantly strengthen our discussion and further underscore the immense potential of our proposed approach. We sincerely appreciate reviewer #3 for encouraging us to provide more

experimental evidence, and we are delighted to present it. By incorporating reviewer #3's valuable insights, we firmly believe that this work is now suitable for publication in *Nature Communications*, offering a broader platform to highlight the important findings presented through our research.

Response fig. 1 | a-b, TEM images of photothermally synthesized (with 15 OD AuBPs, left column) **a**, iron oxide nanoparticles (size ~ 9 nm, SD 0.5 nm) and **b**, silver nanoparticles (Fig. 4b with 80 °C temperature difference compared to conventional heating using 100 mg silver precursor, size ~ 4 nm, SD 0.2 nm), compared with corresponding Sigma Aldrich products (right column). **c-f**, TEM images of palladium nanocubes obtained through **c**, conventional heating (Supplementary Reference 3, size ~ 10 -11 nm, SD 0.3 nm) at 85 °C and **d-e**, photothermal synthesis with 2 OD AuBPs (size ~ 11 -12 nm, SD 0.3 nm) at 60 °C with **f**, corresponding Pd@AuBP nanohybrids (size of Pd ~ 12 -13 nm, SD 0.3 nm).

Reviewer #4 (Remarks to the Author):

The authors tried hard to answer reviewers' questions and revised the paper accordingly.

Response 4-general summary: We thank reviewer #4 for appreciating our efforts to address the previous queries, and genuinely taking interest in our work. The reviewer's follow-up concerns are addressed below.

1. However, I am not convinced that the revised version shows improved quality compared to the original version. Worse, some of the new additions reflect the fact that the synthetic method they presented is of little use as a general method for nanoparticle synthesis. Judging from the synthetic perspective, I do not feel the reported synthesis is an advance in the field as the nanoparticles made from the synthesis have poor size and shape distributions (The authors claim they have monodispersed nanoparticles, but this is not supported by TEM images they presented).

Response 4-1: Unfortunately, we cannot comment on the new additions that supposedly worsened our manuscript as there is no mention to what those additions are. We kindly direct reviewer #4 to response 2 and 3-general summary where we addressed the quality issue. In response to the raised concern, we wish to kindly point reviewer #4's attention to the standard deviations in the sizes of the photothermally synthesized nanoparticles. These standard deviations, as depicted in response fig. 1

and supplementary figures 5, 13, 21, and 49, are all found to be less than 10%. Thus, we don't think it is a stretch for the photothermally synthesized nanoparticles to be defined as monodisperse (response reference 1).

2. In answering the synthetic yield question, the authors did not give the yield numbers, rather, they gave a method how the synthetic yield should be calculated, which is itself questionable. To use ICP-AES to analyze metal concentration from the reaction mixture and then to calculate synthetic yield can be very misleading. The fact that the authors cannot provide the real yield numbers suggest that they could not even separate any measurable amount of product from their synthesis started with hundreds of mg of metal precursors. The synthesis with such a low product output has no much use for others in the field.

Response 4-2: There appears to be a significant misunderstanding, we regret that the synthetic yield calculation was not sufficiently clear. We recognize that the statement in the method section regarding the 'withdrawal of aliquots from the reaction mixtures', may have inadvertently contributed to this confusion. Consequently, we have precisely outlined the synthetic yield calculation and provided explicit yield values for clarity (response fig. 2). It is important to mention that we are capable of isolating substantial amounts of product following the detailed isolation and purification steps described in the method section (response fig. 3). Additionally, ICP-OES samples were prepared from purified product, therefore the metal detected surely comes from the isolated NPs. Comparing the amount of metal detected by the ICP with a known amount of precursor surely provides an accurate measure of the reaction yield. When dealing with colloidal nanoparticles, we deliberately refrain from fully drying them, as this could result in aggregation or alterations to their physical and chemical properties. Instead, we store these nanoparticles in appropriate solution-based formulations after purification and utilize them accordingly (in the same way suppliers like Sigma Aldrich sell NPs). Moreover, it is again crucial to recognize that employing a weighing method for yield calculation would include the contribution of ligands present on the surface of the colloidal particles. This approach may result in less precise, qualitative, and potentially erroneous results. To address these combined issues, we adopted a more sophisticated approach by utilizing ICP-OES with detection limits in the 10^{-2} to 10^{-7} mg/kg (ppm) range according to the sensitivity of individual elements. This technique allows us to determine the actual concentrations, considering only the nanoparticles themselves without the interference of stabilizing ligands or surface impurities. We kindly request reviewer #4 to refer to the revised SI, page number 5 and supplementary fig. 31, 39 and 48 for further details.

Response fig. 2 | a-b, Synthetic yield of iron oxide nanoparticles under photothermal (with 15 OD AuBPs) and conventional reactions. **a,** Iron oxide nanoparticle synthesis for **i,** ~4-6 nm particles, **ii,** ~10 nm particles, **iii,** ~6-8 nm triangular particles between 180-250 °C. **b,** Iron oxide nanoparticle synthesis corresponding to **i,** ~2-4 nm particles, and **ii,** ~15 nm nanoplates at 200-300 °C. **c,** Iron oxide nanoparticle synthesis at a photothermal temperature of 200 °C with different concentrations (15-25 OD) of AuBPs with a slightly greater yield at higher AuBP concentrations. **d,** Silver nanoparticle, **e,** silver nanowire, and **f,** palladium nanoparticle synthesis in different reaction conditions exhibiting consistent and greater nanoparticle yield under photothermal conditions (with 15 OD AuBPs for silver, and 2 OD for palladium) at lower temperatures similar to the iron oxide nanoparticles.

Response fig. 3 | a, Purified colloidal iron oxide nanoparticle powder (110 mg, 4-6 nm) synthesized photothermally with 15 OD AuBPs at 180 °C. **b,** Scale-up synthesis showing the capability of producing gram-scale product.

3. The claimed assembly of nanoparticles on SiO₂-coated Au during the synthesis is also not supported. One would not agree to call the random disposition of nanoparticles on the SiO₂ surface a controlled assembly. A much-controlled assembly on SiO₂ can be done easily with well-established silane surface chemistry. Therefore, the claimed assembly method has no advantage to what has been known in the field.

Response 4-3: We sincerely appreciate reviewer #4's thoughtful query regarding the silane surface chemistry-based approach employed to generate these hybrid particles. We have been rigorously working on the development of SiO₂@AuBP-based hybrid structures using the mentioned strategy for an extended period, even prior to conceptualizing the current project. Although our previous attempts yielded limited success, we have presented further clarification in response fig. 4 demonstrating our endeavors to create hybrid structures by combining silane-modified positively or negatively charged SiO₂@AuBPs with negatively or positively charged iron oxide NPs. The resulting hybrids, attained through this process, could be described as random deposition with multiple lengthy steps and lack of precise control over the reaction operando. The photothermal method for the synthesis of the described hybrids yields homogeneously dispersed IONPs on the silica surface. Moreover, the size and concentration of the deposited particles on the surface can be controlled effectively by simply regulating the time of exposure to light. In addition, it is crucial to emphasize that the formation of Pd assemblies is not related to silane chemistry, as previously detailed in our comprehensive response report. Therefore, the utilization of the photothermal approach allows for greater control and more uniform hybrid structures all while streamlining their production. This represents a noteworthy improvement over the time-consuming silane surface-chemistry based method. This discussion is now added in the revised SI as supplementary fig. 23 for the benefit of our readers which in turn further strengthens our current findings (Supplementary fig. 13, 21, and Table 6).

Response fig. 4. **a**, Silane chemistry employed surface modification strategy of nanoparticles. **b**, Change in zeta potential values (positive for AEAPS, negative for hydroxyl) post surface modification of the nanoparticles (reference 44). **c**, TEM images of oppositely charged nanoparticles in an attempt for synthesis of nanoparticle-SiO₂@AuBP hybrids, left, through AuBP(+)/IO(-) and right, through AuBP(-)/IO(+). **d**, TEM images of nanoparticle-SiO₂@AuBP hybrids obtained through photothermal reaction (fig. 2b) with control of number of iron oxide (IO) particles on the gold bipyramids (SiO₂@AuBPs) operando simply via controlling reaction time through light irradiation, left 45 min and right 3 h.

Reviewer #5 (Remarks to the Author):

The authors addressed my concerns adequately.

Response 5-general summary: We sincerely thank reviewer #5 for the kind appreciation of our diligent efforts in addressing the previous queries.

Response references.

1. Muzzio, M., Li, J., Yin, Z., Delahunty, I. M., Xie, J & Sun, S. Monodisperse nanoparticles for catalysis and nanomedicine. *Nanoscale*, **11**, 18946-18967 (2019).

Reviewer comments, third round -

Reviewer #1 (Remarks to the Author):

Questions have been answered correctly.

Reviewer #3 (Remarks to the Author):

Although the authors attempted to answer the reviewer's comments, my opinion on the manuscript remains. I think Nature Communications is a highly competitive journal that demands research investigation showing important advances in the field. The synthesis methodology described by Biswas and coworkers is interesting. However, it does not facilitate the synthesis of colloidal nanoparticles with improved size control (both in size tunability and distribution) or novel morphologies than other methods found in the current literature. The energy efficiency argument is appealing, but I doubt that energy reduction is significant considering the energy, reactants, solvents, materials, and time required to synthesize the plasmonic heaters covered by silica. For these reasons, I do not recommend publication in Nature Communications but instead in a more specialized journal.

We would like to sincerely thank all reviewers for dedicating the time and effort in reviewing this manuscript. We believe that the comments made by all reviewers have given us fundamental insight and helped substantially to improve our work further. Please find below a thorough point-by-point response to all the reviewers' comments and suggestions.

Point-by-point responses of reviewers' comments:

Reviewer #1 (Remarks to the Author):

Questions have been answered correctly.

Response 1: We would like to express our appreciation for reviewer #1's positive feedback and recognition of our revisions to previous comments.

Reviewer #3 (Remarks to the Author):

Although the authors attempted to answer the reviewer's comments, my opinion on the manuscript remains. I think Nature Communications is a highly competitive journal that demands research investigation showing important advances in the field. The synthesis methodology described by Biswas and coworkers is interesting. However, it does not facilitate the synthesis of colloidal nanoparticles with improved size control (both in size tunability and distribution) or novel morphologies than other methods found in the current literature. The energy efficiency argument is appealing, but I doubt that energy reduction is significant considering the energy, reactants, solvents, materials, and time required to synthesize the plasmonic heaters covered by silica. For these reasons, I do not recommend publication in Nature Communications but instead in a more specialized journal.

Response 3-general summary: We would like to thank reviewer #3 for finding our work interesting. We believe that we have adequately addressed queries regarding the quality aspects in our previous response letter. We thank the reviewer for taking the time to review our work, we appreciate the reviewer's comments and criticism throughout the reviewing process and have no doubt it has helped us to improve our research.